# Hydrochemical Characteristics and Evolution of Groundwater in the Alluvial Plain (Anqing Section) of the Lower Yangtze River Basin: Multivariate Statistical and Inversion Model Analyses

Qiaohui Che [1,2], Xiaosi Su [1,2,*], Shixiong Wang [3], Shida Zheng [1,2] and Yunfeng Li [4]

1   Institute of Water Resources and Environment, Jilin University, Changchun 130012, China; cheqiaohui@126.com (Q.C.); shidazheng1101@126.com (S.Z.)
2   College of Construction Engineering, Jilin University, Changchun 130021, China
3   Hebei Provincial Prospecting Institute of Hydrogeology and Engineering Geology, Shijiazhuang 050021, China; hbskywsx@163.com
4   Nanjing Geological Survey, China Geological Survey, Nanjing 210016, China; liyf@mail.cgs.gov.cn
*   Correspondence: suxiaosi@163.com

**Abstract:** The alluvial plain (Anqing section) of the lower reaches of the Yangtze River basin is facing increasing groundwater pollution, not only threatening the safety of drinking water for local residents and the sustainable development and utilization of groundwater resources but also the ecological security of the Yangtze River Basin. Therefore, it is necessary to conduct a preliminary analysis on the hydrochemical characteristics and evolution law of groundwater in this area. This study aimed to evaluate potential hydrogeochemical processes affecting the groundwater quality of this area by analyzing major ions in groundwater samples collected in 2019. Compositional relationships were determined to assess the origin of solutes and confirm the predominant hydrogeochemical processes controlling various ions in groundwater. Moreover, factors influencing groundwater quality were evaluated through the factor analysis method, and the control range of each influencing factor was analyzed using the distribution characteristics of factor scores. Finally, reverse hydrogeochemical simulation was carried out on typical profiles to quantitatively analyze the hydrochemical evolution process along flow paths. The Piper trilinear diagram revealed two prevalent hydrochemical facies, Ca-HCO$_3$ type (phreatic water) and Ca-Na-HCO$_3$ type (confined water) water. Based on the compositional relationships, the ions could be attributed to leaching (dissolution of rock salt, carbonate, and sulfate), evaporation and condensation, and cation exchange. Four influencing factors of phreatic water and confined water were extracted. The results of this study are expected to help understand the hydrochemical characteristics and evolution law of groundwater in the alluvial plain (Anqing section) of the lower Yangtze River basin for effective management and utilization of groundwater resources, and provide basic support for the ecological restoration of the Yangtze River Basin.

**Keywords:** hydrogeochemistry; ionic ratios; factor analysis; inverse modeling; Yangtze River

## 1. Introduction

Groundwater resources are an important constituent of water resources. The temporal and spatial distribution of groundwater quality reflects the formation and evolution characteristics, geological and hydrogeological background, and influencing factors of groundwater, which are hot topics in hydrogeological and hydrogeochemical research [1–4]. An in-depth understanding of the interaction mechanism between groundwater and the environment can be obtained by investigating the spatio–temporal variation characteristics and evolution rules of groundwater hydrochemistry. The chemical composition of groundwater is a multivariable and complex function [5], and its formation and evolution are affected by the characteristics of aquifer media, chemical composition, hydrodynamic conditions, and human factors [6–9]. Therefore, the formation and geochemical evolution of groundwater are complex. Conventionally, various methods have been used for

studying the geochemical evolution of groundwater, mainly including the Piper diagram method [10,11], Gibbs graph method [12], and ion ratio method [13,14]. These methods are often simple and intuitive. Multivariate statistical methods have been used to determine the relationship and influence among multivariate. By extracting the mathematical characteristics of data and ignoring the evolution mechanism of hydrochemical components, water quality factors can be described regionally to study the spatiotemporal distribution of the hydrochemical characteristics of groundwater, evaluate water quality, and identify influencing factors [15–19]. Integrating hydrochemical interpretation with inverse modeling, models with high confidence levels can be applied to quantitatively identify hydrochemical processes along a flow path [20,21]. Inverse geochemical modeling in PHREEQC [22] is based on a geochemical mole-balance model, which calculates phase mole transfer (moles of minerals and gases that must enter or leave a solution) to account for differences in initial and final water compositions along a flow path in a groundwater system. This model requires the input of at least two chemical analyses of groundwater at different points of the flow path and a set of phases (minerals and/or gases) that potentially react along this flow path [23].

The Yangtze River is the largest river in China and the third-largest river in the world. It plays an important role in the sustainable development of the regional economy and ecology [24,25]. To strengthen the protection and restoration of the ecological environment in the Yangtze River basin, facilitate the effective and rational use of resources, safeguard ecological security, ensure harmony between humans and nature, and achieve the sustainable development of the Chinese nation, the 24th Standing Committee session of the 13th National People's Congress passed the first river basin law "Yangtze River Protection Law" on 26 December 2020, and this law came into effect on 1 March 2021. Anqing is located beside the Yangtze River on the alluvial plain in the lower reaches. It is an important city in the Yangtze River Economic Belt and the Yangtze River Delta. Since the 1980s, many large-scale chemical plants have been built in Anqing, posing a serious risk of water pollution [26–28]. Therefore, the study of the interaction between surface water and groundwater in Anqing is of great practical significance to the prevention and control of water pollution and the restoration of the ecological environment in the Yangtze River.

In this study, the main controls on groundwater hydrogeochemistry and hydrochemical characteristics in the alluvial plain (Anqing) in the lower reaches of Yangtze River Basin were analyzed using the Piper diagram, ion ratio, and statistical analysis methods. A reverse hydrogeochemical simulation was also performed to quantitatively analyze the evolution process of groundwater along the groundwater flow path in certain areas. Detailed information on hydrogeochemical mechanisms affecting the concentrations and distributions of dissolved ions in complex geological and hydrogeological systems would provide a scientific basis for better groundwater resource development and management at the local scale and the restoration of the ecological environment of the Yangtze River Basin.

## 2. Study Area

Anqing is located in southeastern China, on the north bank of the lower reaches of the Yangtze River. It lies between $29°47'$–$31°16'$ N and $115°45'$–$117°44'$ E, spanning three geomorphic units: the middle and low mountains of the Dabie Mountains, low hills along the Yangtze River, and alluvial plain along the Yangtze River. The topography has a general trend of higher in the northwest and lower in the southeast. The Dabie Mountains have an altitude of more than 400 m a.s.l. in the northwest and 100–200 m a.s.l. in the middle; the alluvial plain of the Yangtze River is flat in the south. The alluvial plain of the Yangtze River was taken as the study area (Figure 1).

Anqing is located in the northern subtropical humid climate zone, with a mild climate and moderate rainfall. The annual average temperature ranges from 14.4 °C to 16.6 °C, with obvious geomorphic zonation. The annual average temperature in the Dabie Mountain area is 14.4 °C, and that in the area along the Yangtze River is 16.1 °C to 16.6 °C. The multi-year average rainfall is 1466.2 mm, and the multi-year average evaporation is 917.4 mm.

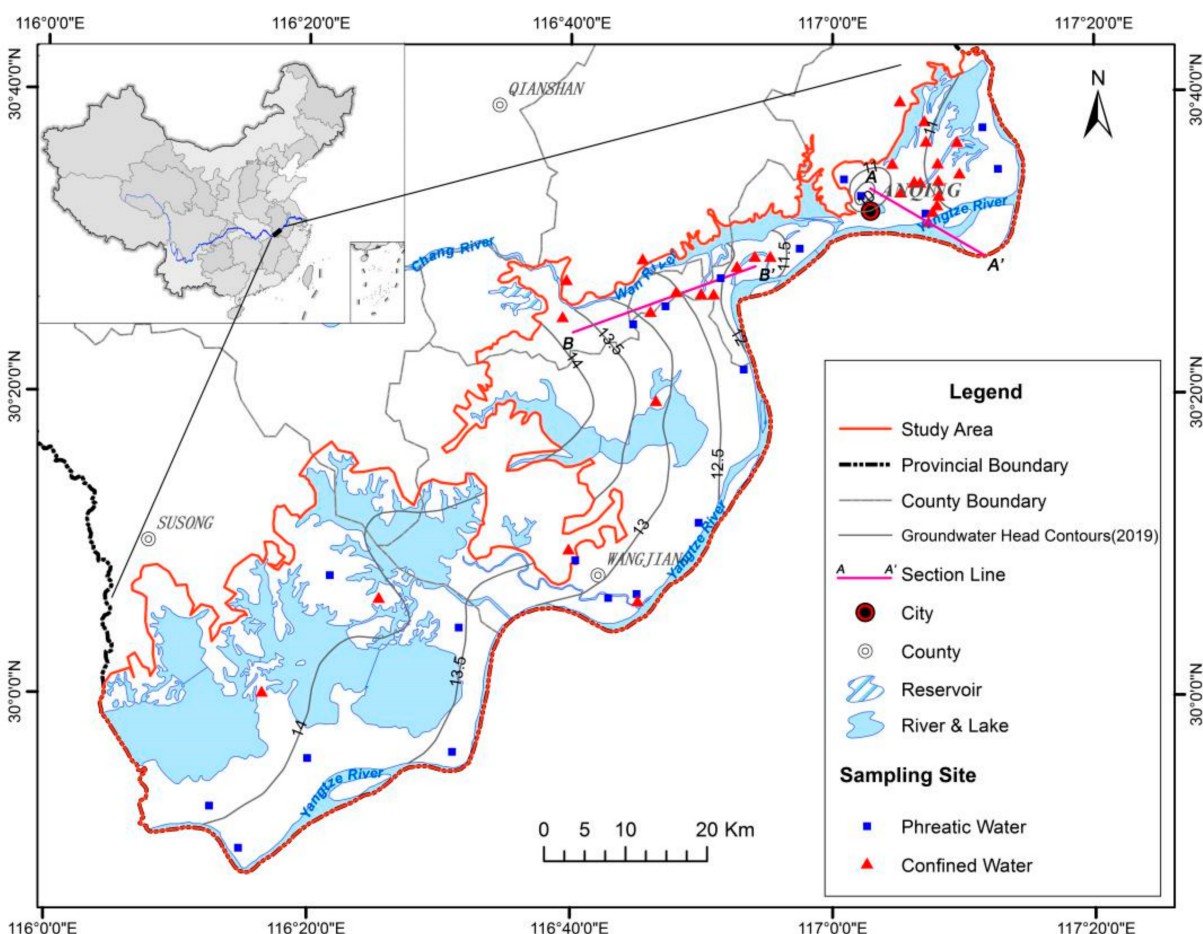

**Figure 1.** Alluvial plain (Anqing section) of the lower Yangtze River basin and sampling locations.

The study area has a well-developed surface water system, with many rivers and lakes. In the study area, the main stream of the Yangtze River is approximately 243 km long, and the Wan River, with a total length of 94 km, is its primary tributary. An obvious peak of river water level is observed every year. The lowest and highest water levels are observed during January–February and July–August, respectively. According to the water level of the Yangtze River monitored at the Anqing hydrological station for the period 2009–2018, the highest water level is 16.98 m (July 2016) and the lowest is 5.72 m (February 2014). The main lakes include Longgan Lake, Daguan Lake, Po Lake, and Pogang Lake.

Quaternary strata in the study area are well developed and distributed from the lower Pleistocene to Holocene. The gravel layer of the lower Pleistocene Anqing Formation is partly exposed in the third terrace and partly buried in the lower part of the second terrace. The gravel layer has a thickness of 15–30 m and unconformably overlies the Red Bed basement. The gravel is mainly composed of quartzite and quartz sandstone, with good sorting and roundness, and the particle size can reach 1–6 cm. The lower part of the Middle Pleistocene Qijiaji Formation is a 1–4 m thick mud-bearing gravel layer, and the upper part is a 3–8 m thick reticulated laterite. The lower member of the upper Pleistocene Xiashu Formation is a 3–6 m thick khaki sub-clay, containing iron and manganese, widely distributed in the second terrace; the upper member is a light yellow sub-clay, mainly distributed in the first terrace. The stratum of the Holocene Wuhu Formation is mainly distributed in the alluvial plains of the Yangtze River and the main tributary valleys of the Wan River. The stratum can be divided into 3 sections from bottom to top: the lower part comprises a gravel layer and gravel-bearing medium-coarse sand (approximately 10 m thick); the middle part comprises medium–fine sand (10–20 m thick); the upper part

comprises grayish yellow–blue gray silty clay (4–10 m thick). In the area with fluvial–lacustrine sediments, the Wuhu Formation is deposited only on the shallow surface, with a thickness not more than 3 m.

Quaternary aquifers in the study area are mainly Holocene sand and gravel phreatic aquifers and lower Pleistocene gravel confined aquifers, with thicknesses of 7–50 m and 0–24 m, respectively. There is no continuous aquitard between the aquifers, but some areas have relative aquitards, as shown in Figures 2 and 3.

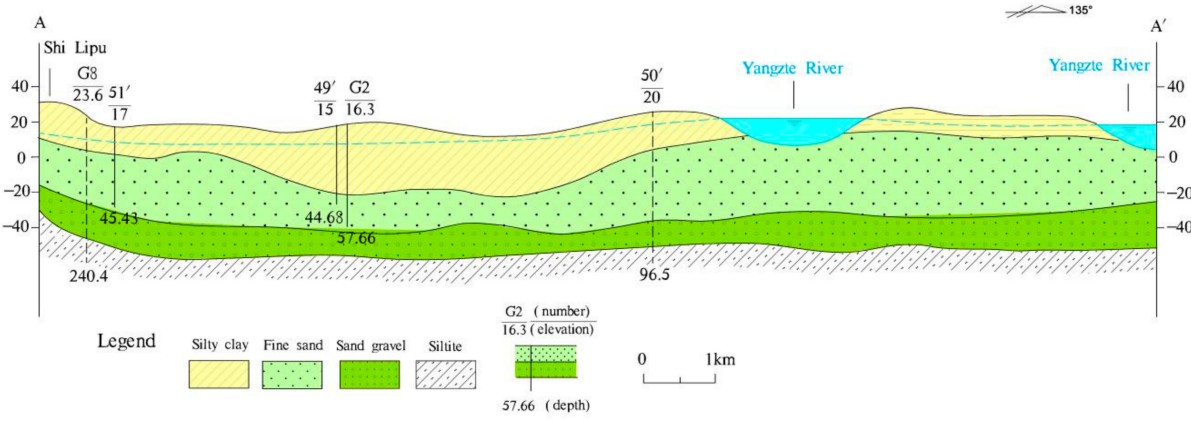

**Figure 2.** Hydrogeologic cross sections along the A–A' transect in Figure 1.

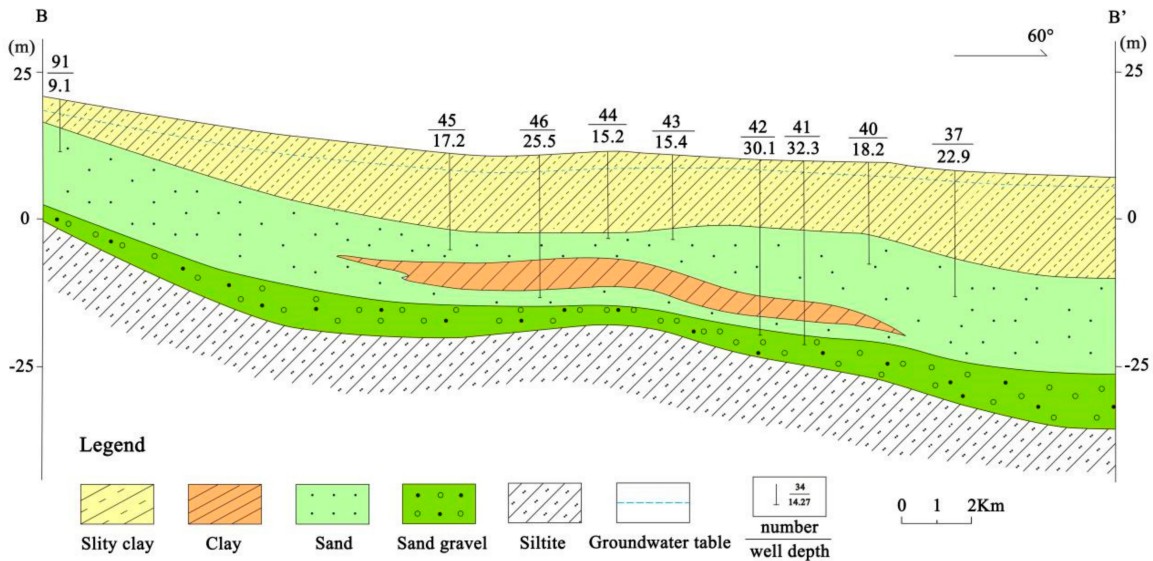

**Figure 3.** Hydrogeologic cross sections along the B–B' transect in Figure 1.

In the western part of the study area, groundwater flows from the piedmont to the Yangtze River and receives lateral recharge from piedmont groundwater. Groundwater runoff occurs in the low mountain and hilly plain area with a small hydraulic gradient, discharging to surface water bodies such as lakes and rivers in the runoff path. Finally, the groundwater flows through the plain area along the Yangtze River and drains into the Yangtze River. However, under the influence of large-scale exploitation of groundwater in urban industrial areas, river water is artificially stimulated to recharge groundwater in the plain along the Anqing urban area. The contour of the groundwater level in 2019 is shown in Figure 1.

## 3. Materials and Methods

### 3.1. Sample Collection and Analysis

In this study, groundwater samples were collected in May 2019, including 20 groups of phreatic water samples and 31 groups of confined water samples. The distribution of sampling points is shown in Figure 1.

Before sampling, containers were soaked in 10% nitric acid solution for 1–2 days, then in tap water for 1–2 days, and rinsed to neutrality. Finally, they were washed with demineralized water three times and then dried at 70 °C on standby.

During sample collection, water was pumped for more than five minutes to discharge long-term residual groundwater in the well pipe. A HACH water quality rapid detector was used to measure the water temperature (T), TDS, dissolved oxygen (DO), conductivity, pH, and oxidation-reduction potential (Eh) of the water samples. The water samples were collected after the readings stabilized. Each sample was collected after rinsing the container with the water sample more than 3 times.

The concentrations of Fe and Mn were determined in the field using Hach DR1900 portable spectrophotometer. As concentration was determined by inductively coupled plasma mass spectrometry (ICP-MS, Agilent 7500C, Santa Clara, CA, USA) and hydride generation atomic fluorescence photometry (AFS 9600, Beijinghaiguang). Anions and cations were analyzed by ion chromatography (881 compact IC, Metrohm, Switzerland). The mass concentration of bicarbonate ion ($HCO_3^-$) was determined by acid-base titration in the laboratory. The reliability of water sample data was tested using the anion and cation balance test method, and the absolute value of the relative error of the anion and cation balance less than 5% was taken as reliable.

### 3.2. Qualitative Research Method

### 3.2.1. Ionic Ratios

In the process of groundwater circulation, the regularity of each ion component and some ion ratios will change. Therefore, the characteristics of ionic combinations and related ion ratios in groundwater can be used to assess the genesis of groundwater, and identify the source of the chemical components of groundwater and mixing process of different water bodies; this is an effective approach for analyzing the evolution of groundwater [29–31]. However, the variation of ion concentration is largely affected by mixing. When the ion composition is near the mixing line, it indicates that there is no water–rock interaction. When the ion composition deviates from the mixing line, it indicates that it is affected by water–rock interaction. The millimolar per liter (mmol/L) ratio between ($Na^+ + K^+$) and $Cl^-$ ($\gamma(Na+K)/\gamma Cl$ can reflect the source of $Na^+$ and $K^+$. A $\gamma(Na+K)/\gamma Cl$ close to 1 indicates the dissolution of halite; a $\gamma(Na+K)/\gamma Cl$ ratio greater than 1 indicates the dissolution of silicates or cation exchange. The main sources of $Ca^{2+}$ and $Mg^{2+}$ in groundwater are mainly the dissolution of carbonates or silicates and evaporites. Accordingly, the mmol/L ratio between $Ca^{2+}$ and $SO_4^{2-}$ ($\gamma Ca^{2+}/\gamma SO_4^{2-}$, milliequivalents per liter (meq/L) ratios between $Ca^{2+}$ and $HCO_3^-$ ($\gamma Ca^{2+}/\gamma HCO_3^-$) and $Ca^{2+} + Mg^{2+}$ and $HCO_3^-$ ($\gamma(Ca^{2+}+Mg^{2+})/\gamma HCO_3^-$) can be used to determine the main sources of $Ca^{2+}$ and $Mg^{2+}$. A $\gamma Ca^{2+}/\gamma SO_4^{2-}$ ratio close to 1, corresponds to gypsum dissolution. A $\gamma Ca^{2+}/\gamma HCO_3^-$ ratio close to 1, corresponds to calcite dissolution. A $\gamma(Ca^{2+}+Mg^{2+})/\gamma HCO_3^-$ close to 1, corresponds to dolomite dissolution. The meq/L ratio of ($SO_4^{2-}+Cl^-$) to $HCO_3^-$ ($\gamma(SO_4^{2-}+Cl^-)/\gamma HCO_3^-$) reflects the main source of chemical components in groundwater. A $\gamma(SO_4^{2-}+Cl^-)/\gamma HCO_3^-$ ratio greater than 1 indicates evaporite dissolution as the main contributor to the chemical composition of groundwater. A $\gamma(SO_4^{2-}+Cl^-)/\gamma HCO_3^-$ ratio less than 1 indicates carbonate dissolution as the main contributor to the chemical composition of groundwater. The ratio of $\gamma(Na^++K^+-Cl^-)/\gamma(Ca^{2+}+Mg^{2+}-HCO_3^--SO_4^{2-})$ can be used to reflect cation exchange. In the presence of cation exchange, $\gamma(Na^++K^+-Cl^-)$ will be negatively correlated to $\gamma(Ca^{2+}+Mg^{2+}-HCO_3^--SO_4^{2-})$ with a slope of −1, that is, the content of $Ca^{2+}+Mg^{2+}$ decreases with increasing $Na^++K^+$ content [32–34].

### 3.2.2. Factor Analysis

Factor analysis [35] is a multivariate statistical analysis method with dimensionality reduction. In other words, more samples or variables are replaced by fewer principal factors, which reflect as much information as possible; moreover, the principal factors are independent of each other [36,37]. According to varying research objectives, factor analysis can be divided into the Q type (correlation between samples) and R type (correlation between variables). The basic idea of R-type factor analysis is to group variables according to the correlation, such that the correlation between variables in the same group is higher, but the correlation between variables in different groups is lower. Each set of variables represents a basic structure, namely a factor, which can reflect the observed correlation. In the field of hydrogeochemistry, R-type factor analysis can eliminate independent and repetitive hydrochemical components and summarize numerous intricately interrelated variables to a few common factors. Each main factor represents a basic combination of hydrochemical components. It often indicates the origin of hydrochemical characteristics and can be used to explain complicated relationships between hydrochemical components [38–42].

### 3.3. Quantitative Analysis
Inverse Modeling

PHREEQC is undoubtedly the most widely used reverse hydrogeochemical simulation in the world. In this study, PHREEQC version 3 was used for reverse hydrogeochemical simulation. On the representative flow path, according to the change of sample ion concentration, possible mineral phases in the medium are ascertained, the mineral saturation index and dissolved precipitation of the mineral phase are calculated, and the formation and evolution law of regional groundwater are revealed [43–45].

## 4. Results and Discussion
### 4.1. Hydrochemical Characteristics of Groundwater

The TDS value of phreatic water in the study area ranged from 176.30 to 575.45 mg·L$^{-1}$ with a mean value of 365.42 mg·L$^{-1}$, indicating that the groundwater is fresh water. The pH value ranged from 6.78 to 7.88 with a mean value of 7.39, indicating a weakly alkaline environment. The order of relative abundance of major cations in the groundwater followed $Ca^{2+} > Na^+ > Mg^{2+} > K^+$, and the corresponding average mass concentrations were 1.262 mmol·L$^{-1}$, 2.019 mmol·L$^{-1}$, 0.904 mmol·L$^{-1}$, and 0.079 mmol·L$^{-1}$, respectively. The order of relative abundance of major anions in the groundwater followed $HCO_3^- > Cl^- > SO_4^{2-} > NO_3^-$, and the corresponding average mass concentrations were 5.058 mmol·L$^{-1}$, 0.646 mmol·L$^{-1}$, 0.423 mmol·L$^{-1}$, and 0.214 mmol·L$^{-1}$, respectively. The dominant cations were $Ca^{2+}$ and $Na^+$, and the dominant anions were $HCO_3^-$ in phreatic water. Table 1 shows that the variation coefficients of mass concentrations of Fe, Mn, and As in phreatic water were all greater than 100%, indicating that they are more sensitive and unstable to external inputs, such as hydrological conditions, topography, and human activities. The mass concentrations of Fe, Mn, and As were 0.000–0.427 mmol·L$^{-1}$, 0.002–0.065 mmol·L$^{-1}$, and 0.000–0.165 μmol·L$^{-1}$, respectively. The contents of Fe, Mn, and As in some areas exceeded the WHO drinking water quality standard [46], which stipulates mass concentrations of Fe ≤ 0.3 mg·L$^{-1}$, Mn ≤ 0.1 mg·L$^{-1}$, and As ≤ 10 μg·L$^{-1}$. The chemical groundwater types of the study area were distinguished and grouped by their position on a Piper diagram (Figure 4). Based on the major cation and anion, the following two major hydrochemical facies were identified: $Ca-HCO_3$ and $Ca-Na-HCO_3$ types.

Previous isotopic studies confirmed that phreatic water in the study area is mainly recharged by lake water and rainfall [47], Therefore, the hydrochemistry can be safely presumed to be affected by mixing. The ion concentration of the sample after mixing was calculated based on the oxygen stable isotope $^{18}O$ and compared with the measured data. In this manner, the influence of water-rock interaction on each ion component was determined. In Table 1, the values in bold indicate increases in ion concentration due to water-rock interaction. The specific calculation process is shown in Table S1. The

concentrations of $HCO_3^-$, $Na^+$, $Ca^{2+}$, and $Mg^{2+}$ components were higher than their mixed concentrations, indicating that water–rock interaction generally leads to the dissolution of minerals containing C, Na, Ca, and Mg. The concentrations of $Cl^-$ and $NO_3^-$ also increased in most cases, which is related to halite dissolution and human activities in some areas. $K^+$ concentration changed slightly, indicating that minerals containing K are in equilibrium.

**Table 1.** Statistical characteristics of the chemical composition of phreatic water.

| Sample ID | $\rho(Cl^-)$ | $\rho(NO_3^-)$ | $\rho(SO_4^{2-})$ | $\rho(HCO_3^-)$ | $\rho(Na^+)$ | $\rho(K^+)$ | $\rho(Ca^{2+})$ | $\rho(Mg^{2+})$ | $\rho(TDS)$ | $\rho(Fe)$ | $\rho(Mn)$ | $\rho(As)$ |
|---|---|---|---|---|---|---|---|---|---|---|---|---|
| 11 | 0.287 | 0.041 | 0.279 | 4.227 | **1.371** | 0.014 | **1.425** | 0.277 | 265.640 | $1.071 \times 10^{-3}$ | $3.636 \times 10^{-3}$ | $4.698 \times 10^{-2}$ |
| 12 | **0.676** | **0.070** | **0.988** | **5.566** | **2.132** | **0.077** | **1.683** | **1.103** | 428.950 | $3.571 \times 10^{-4}$ | $1.818 \times 10^{-3}$ | 0.000 |
| 16 | **1.156** | **0.158** | **0.478** | **4.686** | **3.339** | **0.069** | **1.403** | **0.862** | 377.020 | $7.143 \times 10^{-4}$ | $1.818 \times 10^{-3}$ | $1.411 \times 10^{-2}$ |
| 18 | **0.770** | **0.329** | **0.539** | **5.021** | **2.716** | **0.064** | **1.458** | **1.130** | 411.650 | $7.143 \times 10^{-4}$ | $1.818 \times 10^{-3}$ | 0.000 |
| 48 | **0.554** | 0.000 | **0.560** | **5.394** | **1.827** | **0.028** | **1.571** | **1.155** | 423.590 | $3.036 \times 10^{-3}$ | $5.455 \times 10^{-3}$ | 0.000 |
| 52 | 0.369 | **0.244** | 0.277 | 2.011 | 1.056 | 0.025 | **1.211** | 0.177 | 176.300 | $1.250 \times 10^{-3}$ | $7.273 \times 10^{-3}$ | $3.975 \times 10^{-2}$ |
| 54 | 0.365 | **0.324** | 0.302 | 3.682 | 0.882 | **0.079** | 0.926 | 0.786 | 285.660 | $1.071 \times 10^{-3}$ | $3.636 \times 10^{-3}$ | $1.262 \times 10^{-2}$ |
| 55 | **1.001** | **0.208** | **0.623** | **6.943** | **2.123** | 0.049 | **1.753** | **1.103** | 491.930 | $5.357 \times 10^{-4}$ | $3.636 \times 10^{-3}$ | $2.080 \times 10^{-5}$ |
| 58 | 0.493 | **0.273** | **0.343** | 4.150 | **1.433** | **0.190** | **1.244** | 0.798 | 320.890 | $3.750 \times 10^{-3}$ | $7.273 \times 10^{-3}$ | $1.127 \times 10^{-4}$ |
| 60 | **0.586** | **0.622** | **0.476** | **4.361** | **2.983** | **0.078** | **1.227** | 0.848 | 400.650 | $5.357 \times 10^{-4}$ | $5.455 \times 10^{-3}$ | $6.623 \times 10^{-3}$ |
| 62 | **1.120** | **0.621** | **0.342** | **6.512** | **2.369** | 0.003 | 0.683 | **1.232** | 508.890 | $5.357 \times 10^{-4}$ | $1.818 \times 10^{-3}$ | 0.000 |
| 68 | **0.466** | **0.432** | **0.488** | 1.999 | **1.678** | **0.290** | 0.667 | 0.634 | 253.540 | $2.500 \times 10^{-3}$ | $7.273 \times 10^{-3}$ | $2.223 \times 10^{-2}$ |
| 69 | **0.528** | **0.102** | **0.465** | **8.138** | **1.776** | **0.254** | **1.764** | **1.157** | 393.670 | $1.571 \times 10^{-2}$ | $2.727 \times 10^{-2}$ | $5.066 \times 10^{-2}$ |
| 70 | **0.699** | **0.437** | **0.499** | **8.176** | **2.579** | 0.003 | **1.835** | **1.232** | 524.420 | $1.071 \times 10^{-3}$ | $1.091 \times 10^{-2}$ | $1.404 \times 10^{-4}$ |
| 71 | **0.717** | **0.054** | **0.498** | 2.967 | **2.209** | 0.041 | 0.767 | **0.921** | 245.060 | $7.143 \times 10^{-4}$ | $5.455 \times 10^{-3}$ | 0.000 |
| 73 | 0.344 | **0.125** | **0.250** | **4.724** | 1.055 | **0.069** | **1.294** | 0.656 | 282.700 | $1.071 \times 10^{-3}$ | $3.636 \times 10^{-3}$ | $8.819 \times 10^{-3}$ |
| 74 | **1.115** | **0.077** | 0.207 | 2.630 | **2.438** | 0.014 | 0.135 | 0.875 | 249.460 | $7.143 \times 10^{-4}$ | $1.818 \times 10^{-3}$ | $1.373 \times 10^{-2}$ |
| 78 | **0.544** | 0.037 | **0.362** | **7.727** | **1.985** | 0.000 | **1.699** | **1.114** | 450.080 | $6.071 \times 10^{-3}$ | $5.455 \times 10^{-3}$ | $6.800 \times 10^{-3}$ |
| 83 | **0.602** | 0.000 | 0.137 | **9.640** | **2.527** | 0.006 | **1.902** | **1.481** | 575.450 | $4.268 \times 10^{-1}$ | $6.545 \times 10^{-2}$ | $1.605 \times 10^{-1}$ |
| 86 | **0.531** | **0.120** | **0.354** | 2.611 | **1.901** | **0.224** | 0.595 | 0.547 | 242.887 | $1.250 \times 10^{-3}$ | $7.273 \times 10^{-3}$ | $9.173 \times 10^{-2}$ |
| Minimum | 0.287 | 0.000 | 0.137 | 1.999 | 0.882 | 0.000 | 0.135 | 0.177 | 176.300 | 0.000 | 0.002 | 0.000 |
| Maximum | 1.156 | 0.622 | 0.988 | 9.640 | 3.339 | 0.290 | 1.902 | 1.481 | 575.450 | 0.427 | 0.065 | 0.161 |
| Mean | 0.646 | 0.214 | 0.423 | 5.058 | 2.019 | 0.079 | 1.262 | 0.904 | 365.422 | 0.023 | 0.009 | 0.024 |
| S.D. | 0.265 | 0.193 | 0.185 | 2.205 | 0.658 | 0.088 | 0.489 | 0.330 | 112.684 | 0.095 | 0.014 | 0.040 |
| C.V. | 0.411 | 0.904 | 0.437 | 0.436 | 0.326 | 1.121 | 0.388 | 0.364 | 0.308 | 4.047 | 1.620 | 1.687 |

S.D. stands for standard deviation, C.V. stands for coefficient of variation. Except for As and TDS, which are in $\mu mol \cdot L^{-1}$ and mg/L, respectively, the mass concentrations of other ions and indicators are in $mmol \cdot L^{-1}$. The values in bold indicate increases in ion concentration due to water-rock interaction.

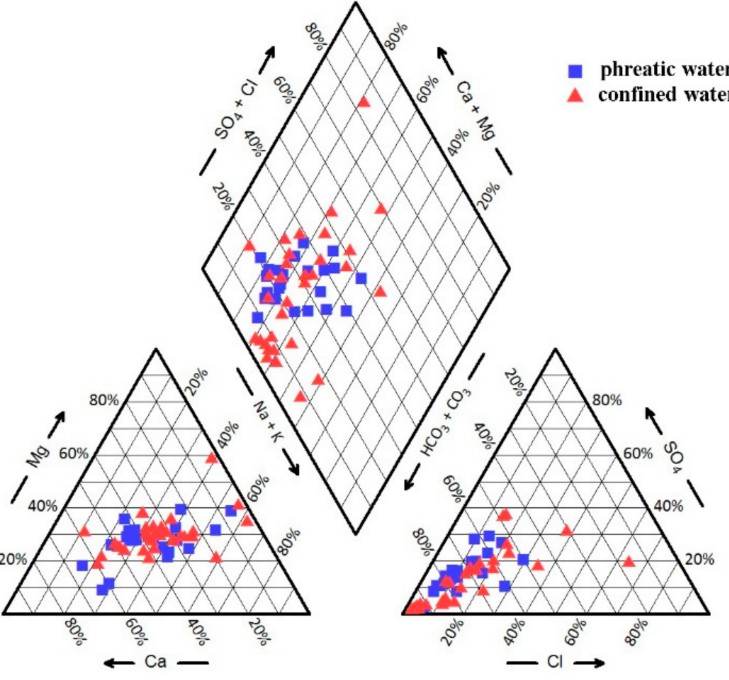

**Figure 4.** Piper diagram of groundwater samples in the study area.

The TDS values of confined water in the study area ranged from 104.12 to 800.00 mg·L$^{-1}$ with a mean value of 363.25 mg·L$^{-1}$, indicating that the groundwater is fresh water. The pH value ranged from 6.46 to 8.47 with a mean value of 7.35, indicating a weakly alkaline environment. The order of relative abundance of major cations in the groundwater followed $Na^+ > Ca^{2+} > Mg^{2+} > K^+$, and the corresponding average mass concentrations were 2.363 mmol·L$^{-1}$, 1.290 mmol·L$^{-1}$, 1.043 mmol·L$^{-1}$, and 0.026 mmol·L$^{-1}$, respectively. The order of relative abundance of major anions in the groundwater followed $HCO_3^- > Cl^- > SO_4^{2-} > NO_3^-$, and the corresponding average mass concentrations were 5.106 mmol·L$^{-1}$, 0.727 mmol·L$^{-1}$, 0.313 mmol·L$^{-1}$, and 0.251 mmol·L$^{-1}$, respectively. In confined water, the dominant cations were $Na^+$ and $Ca^{2+}$ and the dominant anions were $HCO_3^-$. As shown in Table 2, the variation coefficients of mass concentrations of Fe, Mn, and As in confined water were high, indicating that they are more sensitive and unstable to external inputs, such as hydrological conditions, topography, and human activities. The mass concentrations of Fe, Mn, and As were 0.000–0.629 mmol·L$^{-1}$, 0.000–0.060 mmol·L$^{-1}$, and 0.00–1.254 μmol·L$^{-1}$, respectively. The contents of Fe, Mn, and As in some areas exceeded the WHO drinking water quality standard [48]. According to the Piper diagram (Figure 4), the hydrochemical types of confined water are mainly Ca-Na-HCO$_3$, Ca-HCO$_3$, and Ca-Na-HCO$_3$-Cl types.

### 4.2. Qualitative Analysis of the Hydrochemical Evolution Mechanism

### 4.2.1. Ionic Ratios

Using the Mg equivalent ratio relationship between different ions, the water–rock interaction affecting the change of ion concentration can be assessed. However, changes in ion concentration are strongly affected by mixing. In previous studies, we found that lake and precipitation are the main recharge sources of groundwater in the study area. The groundwater mixing line can be obtained by using the ion concentrations of these two end elements (Table S1), as shown in Figure 5.

In terms of the $\gamma(Na+K)/\gamma Cl$ (Figure 5a), the ionic composition of (Na + K) and Cl deviates from the mixing line, indicating that they are affected by water-rock interaction. Moreover, most of the sample points are distributed in the upper left of the 1:1 line. In other words, the mmol/L concentration of $(Na^++K^+)$ is basically greater than that of $Cl^-$. This suggests that $Na^+$ and $K^+$ in groundwater are mainly attributable to halite dissolution, and cation exchange occurs during runoff, resulting in the higher mmol/L concentration of $Na^+$ and $K^+$ ions. In addition, other silicate minerals containing Na and K may also be dissolved.

The $\gamma(SO_4+Cl)/\gamma HCO_3$ (Figure 5b) shows that the ionic composition of (Na + K) and Cl deviates from the mixing line, indicating that they are affected by water-rock interaction. Moreover, the sample points are all distributed below the 1:1 line, and the meq/L concentration of $HCO_3^-$ is much larger than that of $SO_4+Cl$, indicating the dominance of carbonate dissolution.

The $\gamma Ca/\gamma HCO_3$ meq/L ratio relationship (Figure 5c) is consistent with the $\gamma(Na + K)/\gamma Cl$ meq/L ratio relationship; both deviate from the mixing line and 1:1 line and are located at the upper left of the two lines. These results indicate indicating that Ca and HCO$_3$ are affected by water-rock interaction, and the dissolution of gypsum and other calcium-containing minerals may occur in addition to calcite dissolution. The $\gamma Ca/\gamma SO_4$ mmol/L ratio relationship (Figure 5d) shows that the sampling points are distributed between the 1:1 line and the mixing line, indicating that Ca and SO$_4$ are not only affected by mixing but also affected by gypsum dissolution, although only to a small extent.

The $\gamma(Ca + Mg)/\gamma HCO_3$ is shown in Figure 5e. The sampling points are distributed near the 1:1 line and the mixing line, indicating that (Ca + Mg) and HCO$_3$ are jointly affected by mixing and dolomite dissolution.

As indicated by the $\gamma[(Na^++K^+)-Cl^-]/\gamma[(Ca^{2+}+Mg^{2+})(SO_4^{2-}+HCO_3^-)]$ (Figure 5f), the sampling points are generally distributed near the 1:1 line, suggesting a certain degree of cation exchange in groundwater.

**Table 2.** Statistical characteristics of the chemical composition of confined water.

| Sample ID | $\rho(Cl^-)$ | $\rho(NO_3{}^-)$ | $\rho(SO_4{}^{2-})$ | $\rho(HCO_3{}^-)$ | $\rho(Na^+)$ | $\rho(K^+)$ | $\rho(Ca^{2+})$ | $\rho(Mg^{2+})$ | $\rho(TDS)$ | $\rho(Fe)$ | $\rho(Mn)$ | $\rho(As)$ |
|---|---|---|---|---|---|---|---|---|---|---|---|---|
| 6 | 0.670 | 0.513 | 0.345 | 3.959 | 1.381 | 0.038 | 1.935 | 0.602 | 324.840 | $5.357 \times 10^{-4}$ | $1.818 \times 10^{-3}$ | 0.000 |
| 7 | 0.609 | 0.301 | 0.361 | 3.070 | 1.808 | 0.172 | 1.111 | 0.556 | 263.300 | $7.143 \times 10^{-4}$ | $3.636 \times 10^{-3}$ | 0.000 |
| 8 | 1.006 | 0.148 | 0.516 | 5.776 | 2.870 | 0.035 | 1.558 | 1.145 | 430.800 | $1.786 \times 10^{-3}$ | $7.273 \times 10^{-3}$ | 0.000 |
| 9 | 0.800 | 0.537 | 0.326 | 0.857 | 1.456 | 0.023 | 0.368 | 0.453 | 149.610 | $1.250 \times 10^{-3}$ | $3.636 \times 10^{-3}$ | $5.067 \times 10^{-3}$ |
| 19 | 1.764 | 0.000 | 0.836 | 4.906 | 3.065 | 0.003 | 1.735 | 1.308 | 473.580 | $7.143 \times 10^{-4}$ | $7.273 \times 10^{-3}$ | $5.867 \times 10^{-3}$ |
| 20 | 0.809 | 0.140 | 0.400 | 2.974 | 2.113 | 0.003 | 1.096 | 0.700 | 253.930 | $1.071 \times 10^{-3}$ | $5.455 \times 10^{-3}$ | $1.427 \times 10^{-2}$ |
| 21 | 0.655 | 0.000 | 0.166 | 5.346 | 2.779 | 0.000 | 1.433 | 1.181 | 348.560 | $3.571 \times 10^{-4}$ | $1.818 \times 10^{-3}$ | $9.200 \times 10^{-3}$ |
| 22 | 0.512 | 0.162 | 0.287 | 3.663 | 1.647 | 0.000 | 1.095 | 1.168 | 361.920 | $2.679 \times 10^{-3}$ | $5.455 \times 10^{-3}$ | $1.333 \times 10^{-4}$ |
| 23 | 0.441 | 1.762 | 0.322 | 6.070 | 2.723 | 0.000 | 2.897 | 1.528 | 539.930 | $1.071 \times 10^{-3}$ | $3.636 \times 10^{-3}$ | $6.533 \times 10^{-3}$ |
| 25 | 0.804 | 0.563 | 0.970 | 3.548 | 2.378 | 0.053 | 1.468 | 1.156 | 384.970 | $3.571 \times 10^{-4}$ | $1.818 \times 10^{-3}$ | $4.000 \times 10^{-4}$ |
| 36 | 0.880 | 0.037 | 0.507 | 7.364 | 2.894 | 0.026 | 1.795 | 1.010 | 490.310 | $2.500 \times 10^{-3}$ | $1.455 \times 10^{-2}$ | $5.333 \times 10^{-3}$ |
| 37 | 2.494 | 0.383 | 0.480 | 8.186 | 3.014 | 0.019 | 2.885 | 1.475 | 677.650 | $6.964 \times 10^{-3}$ | $1.091 \times 10^{-2}$ | $1.867 \times 10^{-3}$ |
| 39 | 2.058 | 0.000 | 1.286 | 10.806 | 3.859 | 0.049 | 1.819 | 1.591 | 800.000 | $1.232 \times 10^{-2}$ | $2.364 \times 10^{-2}$ | $6.147 \times 10^{-2}$ |
| 40 | 0.235 | 0.000 | 0.076 | 9.860 | 2.889 | 0.000 | 1.759 | 1.528 | 522.450 | $1.813 \times 10^{-1}$ | $6.000 \times 10^{-2}$ | 1.254 |
| 41 | 0.351 | 0.053 | 0.075 | 7.765 | 3.114 | 0.000 | 1.536 | 1.355 | 419.760 | $1.545 \times 10^{-1}$ | $1.636 \times 10^{-2}$ | $2.643 \times 10^{-1}$ |
| 42 | 0.236 | 0.064 | 0.077 | 7.459 | 2.833 | 0.005 | 1.500 | 1.434 | 384.230 | $8.839 \times 10^{-2}$ | $3.818 \times 10^{-2}$ | $1.293 \times 10^{-2}$ |
| 43 | 0.185 | 0.000 | 0.073 | 4.262 | 1.345 | 0.007 | 0.046 | 1.019 | 282.430 | $6.286 \times 10^{-1}$ | $2.364 \times 10^{-2}$ | $7.984 \times 10^{-1}$ |
| 44 | 0.212 | 0.000 | 0.074 | 5.738 | 2.348 | 0.009 | 0.865 | 1.134 | 329.130 | $3.643 \times 10^{-1}$ | $1.818 \times 10^{-2}$ | $9.984 \times 10^{-1}$ |
| 45 | 0.435 | 0.089 | 0.086 | 3.538 | 2.885 | 0.011 | 0.489 | 0.522 | 203.270 | $1.786 \times 10^{-3}$ | $5.455 \times 10^{-3}$ | 0.000 |
| 46 | 0.228 | 0.000 | 0.075 | 4.925 | 2.718 | 0.004 | 0.068 | 0.996 | 277.000 | $1.339 \times 10^{-1}$ | $1.273 \times 10^{-2}$ | $2.803 \times 10^{-1}$ |
| 72 | 0.378 | 0.615 | 0.233 | 1.066 | 1.215 | 0.012 | 0.295 | 0.405 | 104.120 | $2.321 \times 10^{-3}$ | $1.636 \times 10^{-2}$ | $1.333 \times 10^{-4}$ |
| 75 | 0.301 | 0.068 | 0.076 | 4.542 | 2.123 | 0.018 | 0.962 | 0.981 | 259.330 | $7.393 \times 10^{-2}$ | $2.909 \times 10^{-2}$ | $4.787 \times 10^{-2}$ |
| 76 | 0.226 | 0.070 | 0.081 | 5.833 | 3.102 | 0.058 | 1.156 | 1.044 | 311.060 | $7.286 \times 10^{-2}$ | $1.455 \times 10^{-2}$ | $3.907 \times 10^{-2}$ |
| 79 | 1.788 | 0.000 | 0.256 | 9.860 | 1.166 | 0.013 | 2.976 | 1.605 | 644.670 | $2.446 \times 10^{-1}$ | $3.818 \times 10^{-2}$ | $4.125 \times 10^{-1}$ |
| 81 | 0.709 | 0.067 | 0.140 | 5.585 | 1.909 | 0.000 | 1.243 | 1.335 | 245.670 | $1.473 \times 10^{-1}$ | $2.182 \times 10^{-2}$ | $9.867 \times 10^{-3}$ |
| 82 | 0.682 | 0.000 | 0.100 | 4.973 | 3.133 | 0.001 | 0.985 | 1.062 | 310.480 | $1.929 \times 10^{-2}$ | $5.455 \times 10^{-3}$ | $2.680 \times 10^{-2}$ |
| 84 | 0.212 | 0.000 | 0.053 | 6.876 | 3.067 | 0.010 | 1.255 | 1.240 | 361.220 | $3.482 \times 10^{-2}$ | $1.091 \times 10^{-2}$ | $1.023 \times 10^{-1}$ |
| 85 | 0.888 | 1.475 | 0.291 | 4.791 | 2.750 | 0.000 | 2.383 | 1.191 | 463.070 | $4.571 \times 10^{-2}$ | $1.455 \times 10^{-2}$ | $1.173 \times 10^{-2}$ |
| 88 | 1.056 | 0.222 | 0.270 | 1.511 | 2.075 | 0.003 | 0.047 | 0.580 | 207.170 | $7.143 \times 10^{-4}$ | $1.091 \times 10^{-2}$ | $4.000 \times 10^{-3}$ |
| 90 | 0.315 | 0.139 | 0.123 | 0.985 | 0.374 | 0.224 | 0.582 | 0.295 | 197.770 | $5.000 \times 10^{-3}$ | $1.818 \times 10^{-3}$ | $5.467 \times 10^{-3}$ |
| 91 | 0.606 | 0.384 | 0.746 | 2.193 | 2.220 | 0.012 | 0.654 | 0.724 | 238.530 | $3.571 \times 10^{-4}$ | 0.000 | $3.600 \times 10^{-3}$ |
| Minimum | 0.185 | 0.000 | 0.053 | 0.857 | 0.374 | 0.000 | 0.046 | 0.295 | 104.120 | 0.000 | 0.000 | 0.000 |
| Maximum | 2.494 | 1.762 | 1.286 | 10.806 | 3.859 | 0.224 | 2.976 | 1.605 | 800.000 | 0.629 | 0.060 | 1.254 |
| Mean | 0.727 | 0.251 | 0.313 | 5.106 | 2.363 | 0.026 | 1.290 | 1.043 | 363.250 | 0.072 | 0.014 | 0.141 |
| S.D. | 0.578 | 0.413 | 0.298 | 2.592 | 0.775 | 0.049 | 0.801 | 0.377 | 157.786 | 0.135 | 0.013 | 0.312 |
| C.V. | 0.795 | 1.644 | 0.953 | 0.508 | 0.328 | 1.893 | 0.621 | 0.362 | 0.434 | 1.873 | 0.961 | 2.209 |

S.D. stands for standard deviation, C.V. stands for coefficient of variation. Except for As and TDS, which are in $\mu mol \cdot L^{-1}$ and mg/L respectively, the mass concentrations of other ions and indicators are $mmol \cdot L^{-1}$.

In order to further analyze the occurrence and displacement direction of cation exchange in groundwater, chloro-alkaline indices (CAI) were applied [48,49]. The expressions of CAI are as follows:

$$CAI - 1 = \frac{Cl^- - \left(Na^+ + K^+\right)}{Cl^-}$$

$$CAI - 2 = \frac{Cl^- - \left(Na^+ + K^+\right)}{SO_4^{2-} + HCO_3^- + CO_3^{2-} + NO_3^-}$$

The results of CAI-1 and CAI-2 are both negative, indicating the occurrence of cation exchange during runoff and the replacement of $Na^+$ and $K^+$ adsorbed by rocks and soil by $Ca^{2+}$ in groundwater, which is consistent with the $\gamma(Na + K)/\gamma Cl$, $\gamma(Ca + Mg)/\gamma HCO_3$.

### 4.2.2. Factor Analysis

Factor Analysis of Phreatic Water

From the geochemical dataset, principal components were extracted on the symmetrical correlation matrix computed for the 12 variables (Table 3). Before the analysis, the KMO (Kaiser–Meyer–Olkin) and Bartlett (Bartlett test of sphericity) tests were conducted to verify the suitability of the data. The KMO test showed a value of 0.613 and the Bartlett test

showed a significance level of less than 0.01, which indicates that the data have a certain correlation and are suitable for factor analysis.

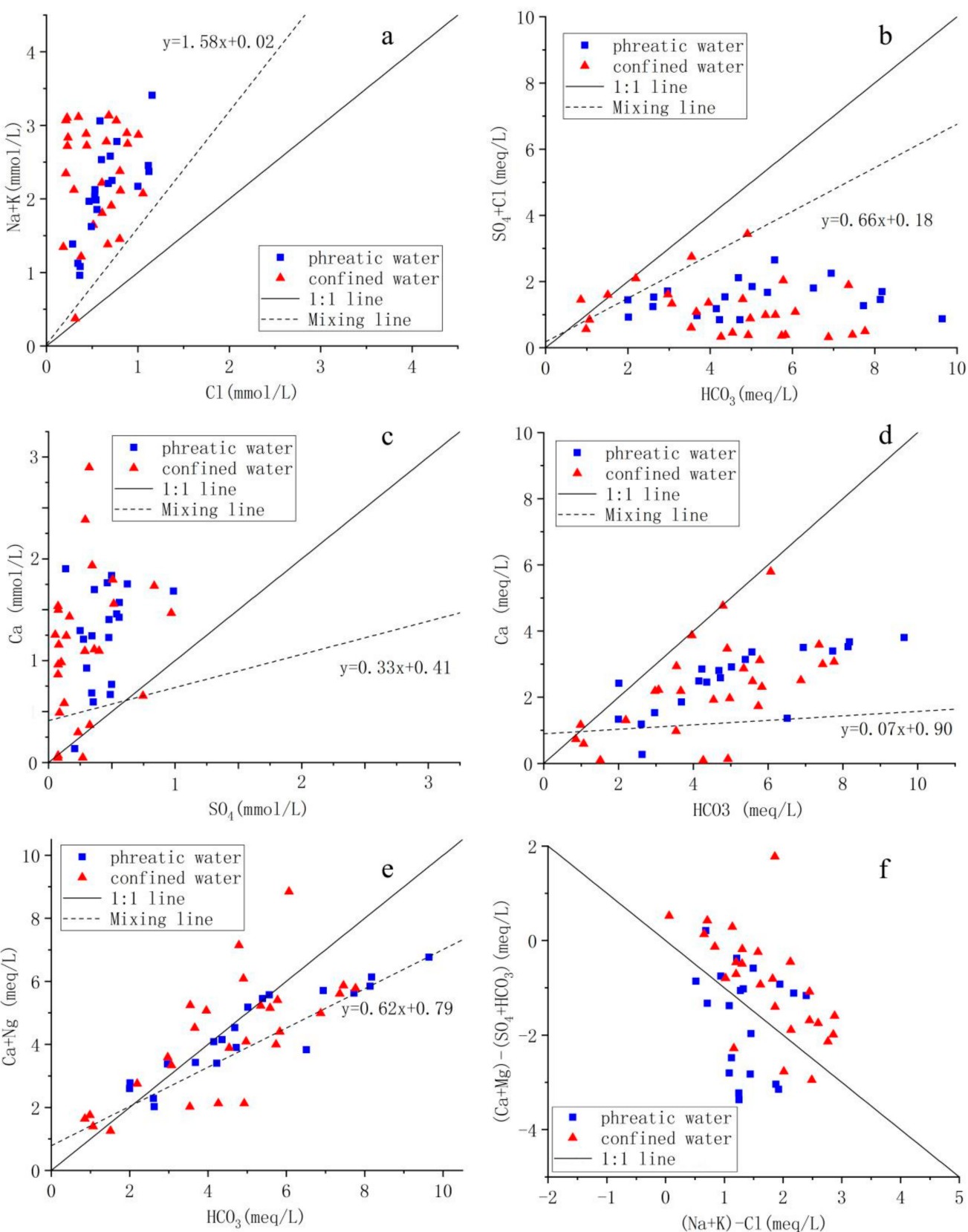

**Figure 5.** Relationships between the rates of the selected ions of groundwater.

**Table 3.** Loading for varimax rotated factor matrix of a four-factor model explaining 80.05% of the total variance.

| Variable | Factor Loading | | | |
|---|---|---|---|---|
| | $F_1$ | $F_2$ | $F_3$ | $F_4$ |
| $Cl^-$ | 0.016 | −0.159 | **0.879** | −0.170 |
| $NO_3^-$ | −0.123 | −0.236 | 0.446 | **0.534** |
| $SO_4^{2-}$ | 0.502 | **−0.638** | 0.073 | 0.240 |
| $HCO_3^-$ | **0.857** | 0.292 | 0.181 | −0.163 |
| $Na^+$ | 0.253 | 0.023 | **0.831** | −0.009 |
| $K^+$ | −0.113 | 0.049 | −0.232 | **0.841** |
| $Ca^{2+}$ | **0.884** | 0.055 | −0.257 | −0.103 |
| $Mg^{2+}$ | **0.729** | 0.150 | 0.542 | −0.114 |
| TDS | **0.825** | 0.150 | 0.465 | −0.096 |
| Fe | 0.311 | **0.892** | 0.084 | −0.133 |
| Mn | 0.393 | **0.900** | −0.018 | 0.025 |
| As | 0.030 | **0.932** | −0.146 | 0.049 |
| Eigenvalue | 3.511 | 3.111 | 2.476 | 1.308 |
| Explained variance% | 27.011 | 23.928 | 19.048 | 10.063 |
| Cumulative% of variance | 27.011 | 50.939 | 69.987 | 80.049 |

Bold values: The maximum absolute value of the loadings of each index.

The main methods of factor load matrix estimation include the principal component method, principal axis factor analysis, and maximum likelihood method. In this study, the principal component method was selected to extract the eigenvalues. Four factors with eigenvalues greater than 1 were selected for analysis, and the cumulative variance contribution rate was 80.05%, indicating that the four factors reflected 80.05% of the information content of the total factors affecting water quality. To highlight typical representative variables of each common factor and explain their practical significance, the factor load matrix was rotated. After rotation, the main factor loads were converted to 1 or 0 polarization. The rotation factor load matrix is shown in Table 3.

$F_1$ reflects water–rock interaction, mainly carbonate dissolution. It was mainly determined by $HCO_3^-$, $Ca^{2+}$, $Mg^{2+}$, and TDS, and its contribution rate was 27.011%. According to the analysis of ion ratios, carbonate dissolution is widely distributed, resulting in the high contents of $HCO_3^-$, $Ca^{2+}$, and $Mg^{2+}$ in groundwater. Figure 6 shows the interpolation of $F_1$ scores at each sampling point of phreatic water. Sampling points with high scores were mainly distributed in the groundwater discharge area (Anqing and Wangjiang sections) and the retention area (Wan River Valley), where groundwater runoff is slow. In these regions, the aquifers have a small grain size, the velocity of groundwater is slow, and water–rock interactions frequently occur between the groundwater and aquifer, resulting in strong carbonate dissolution. These factors contribute to the enhancement of $HCO_3^-$, $Ca^{2+}$, $Mg^{2+}$, and TDS in groundwater.

$F_2$ reflects the endogenous pollution of groundwater, which is affected by aquifer geological conditions. In $F_2$, the factor loads of Fe, Mn, As, and $SO_4^{2-}$ were large, and the contribution rate of $F_2$ was 23.928%. The concentration of Fe and Mn in groundwater in the study area generally exceeds the WHO standard, which can be mainly attributed to the reduction and dissolution of original iron-bearing and manganese-bearing minerals in the aquifer [50]. This is consistent with the geological conditions of the aquifer medium containing iron-bearing and manganese-bearing minerals. With the reduction and dissolution of iron-bearing and manganese-bearing minerals, the content of As in groundwater exceeds the standard. Figure 7 shows the interpolation of $F_2$ scores at each sampling point of phreatic water. Sampling points with high scores almost covered the entire study area, indicating high contents of primary iron-bearing and manganese-bearing minerals in the aquifer medium. In the plain area, the terrain is flat and the groundwater flow rate is slow, which promotes the complete reduction and dissolution of iron-bearing

and manganese-bearing minerals, releasing arsenic in the lattice and affecting groundwater quality. Thus, iron, manganese, and arsenic in groundwater are strongly correlated [51,52].

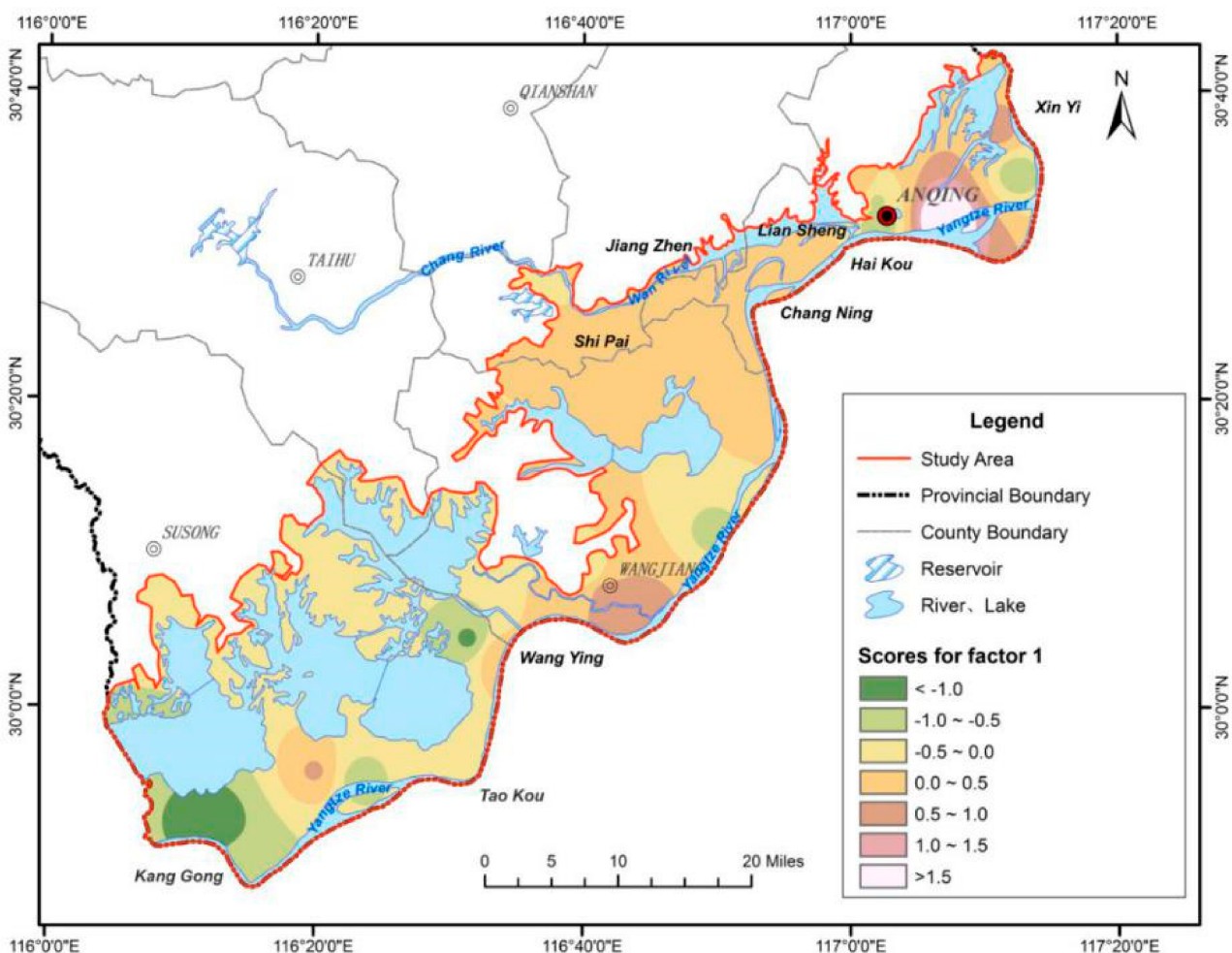

**Figure 6.** Distribution of scores of factor 1 for phreatic water.

$F_3$ reflects the effect of halite dissolution and evaporation-concentration on groundwater hydrochemistry. The loads of $Na^+$ and $Cl^-$ in $F_3$ were large, and the contribution rate of $F_3$ was 19.048%. According to the ion ratio analysis, the chemical composition of phreatic water is affected by evaporation-concentration and halite dissolution in some areas. Figure 8 shows the interpolation of $F_3$ scores at each sampling point of phreatic water. Sampling points with high scores were mainly distributed in the plain along the Yangtze River in the Susong section. The aquifer in this area is shallow, and phreatic water is affected by evaporation concentration. In addition, this area features many lakes, and the groundwater is recharged by lake water, which is affected by evaporation-concentration, resulting in high contents of $Na^+$ and $Cl^-$.

$F_4$ reflects the effect of agricultural production activities on groundwater. In $F_4$, the factor loads of $NO_3^-$ and $K^+$ were large, and the contribution rate of $F_4$ was 10.063%. Figure 9 shows the interpolation of $F_4$ scores at each sampling point of phreatic water. Sampling points with high scores were mainly distributed in the vicinity of Huang Lake and Bo Lake. With a large number of aquaculture farms in this area, fertilizers containing nitrogen and potassium were applied, resulting in the infiltration of $NO_3^-$ and $K^+$ into groundwater with surface water.

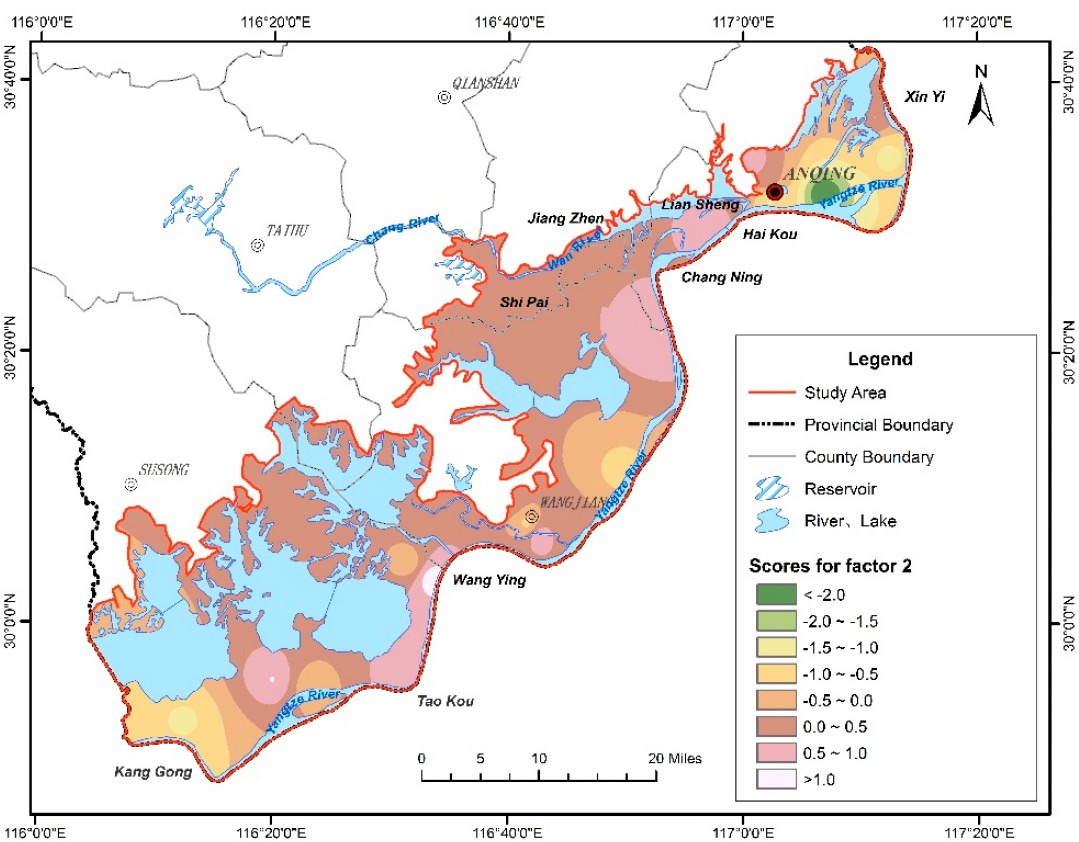

**Figure 7.** Distribution of scores of factor 2 for phreatic water.

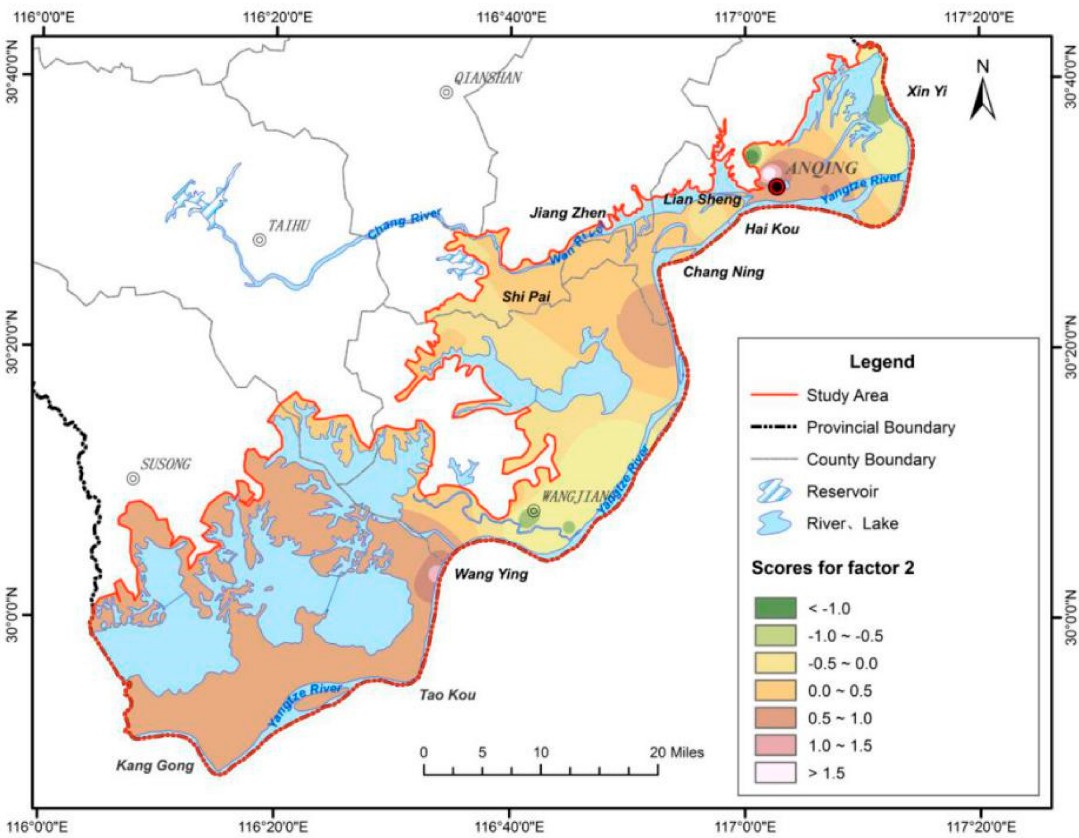

**Figure 8.** Distribution of scores of factor 3 for phreatic water.

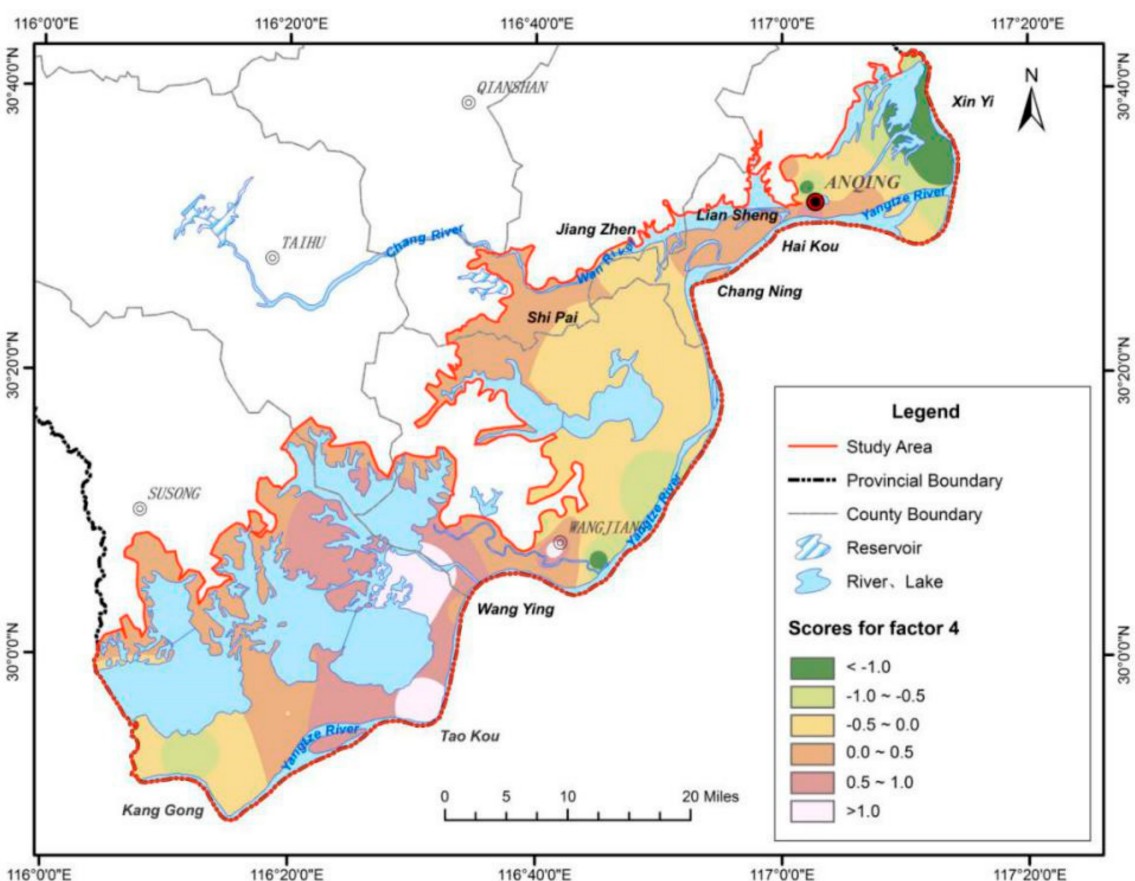

**Figure 9.** Distribution of scores of factor 4 for phreatic water.

Factor analysis of phreatic water revealed that the chemistry of phreatic water in the study area is mainly affected by carbonate dissolution, primary pollution of iron and manganese, halite dissolution, evaporation-concentration, and human activities.

Factor Analysis of Confined Water

Similar to the factor analysis of phreatic water, 12 chemical indexes of confined water samples were selected for analysis. The data were verified for suitability through the KMO and Bartlett tests. The KMO test produced a value of 0.721, and the Bartlett test revealed a significance level of less than 0.01, which indicates that the data have a certain correlation and are suitable for factor analysis.

Four factors with eigenvalues greater than 1 were selected for analysis by the principal component method, and the cumulative variance contribution rate was 79.63%, indicating that the four factors reflected 79.63% of the information content of the total factors affecting water quality. The rotation factor load matrix is shown in Table 4.

$F_1$ reflects water-rock interaction, mainly carbonate dissolution. It was mainly determined by $HCO_3^-$, $Ca^{2+}$, $Mg^{2+}$, and TDS, and its contribution rate was 24.367%. Figure 10 shows the interpolation of $F_1$ scores at each sampling point of confined water. Sampling points with high scores were mainly distributed in the plain along the Yangtze River in the Wangjiang section and the Wan River Valley. The groundwater hydraulic gradient of the riverside plain in the Wangjiang section was relatively large, and groundwater in this area is recharged by groundwater with high contents of $HCO_3^-$, $Ca^{2+}$, and $Mg^{2+}$ from the low mountain and hilly areas. Moreover, the TDS content of phreatic water and confined water in this area is relatively high. In the Wan River Valley plain area, groundwater

runoff is slow, strong water–rock interaction occurs between groundwater and aquifer, and carbonate dissolution is strong, resulting in the high contents of $HCO_3^-$, $Ca^{2+}$, $Mg^{2+}$, and TDS in groundwater.

**Table 4.** Loading for varimax rotated factor matrix of a four-factor model explaining 79.63% of the total variance.

| Variable | Factor Loading | | | |
|---|---|---|---|---|
| | $F_1$ | $F_2$ | $F_3$ | $F_4$ |
| $Cl^-$ | 0.508 | −0.139 | **0.679** | 0.052 |
| $NO_3^-$ | 0.508 | **−0.558** | −0.414 | −0.219 |
| $SO_4^{2-}$ | 0.306 | −0.326 | **0.716** | 0.015 |
| $HCO_3^-$ | **0.619** | 0.511 | 0.219 | 0.478 |
| $Na^+$ | 0.233 | −0.069 | 0.184 | **0.855** |
| $K^+$ | −0.076 | −0.090 | 0.244 | **−0.741** |
| $Ca^{2+}$ | **0.942** | −0.053 | 0.081 | 0.116 |
| $Mg^{2+}$ | **0.690** | 0.369 | 0.061 | 0.532 |
| TDS | **0.831** | 0.210 | 0.379 | 0.283 |
| Fe | −0.035 | **0.816** | −0.272 | −0.080 |
| Mn | 0.240 | **0.796** | −0.059 | 0.173 |
| As | 0.048 | **0.863** | −0.218 | 0.006 |
| Eigenvalue | 3.168 | 2.949 | 2.258 | 1.977 |
| Explained variance% | 24.367 | 22.683 | 17.371 | 15.209 |
| Cumulative% of variance | 24.367 | 47.049 | 64.420 | 79.629 |

Bold values: The maximum absolute value of the loadings of each index.

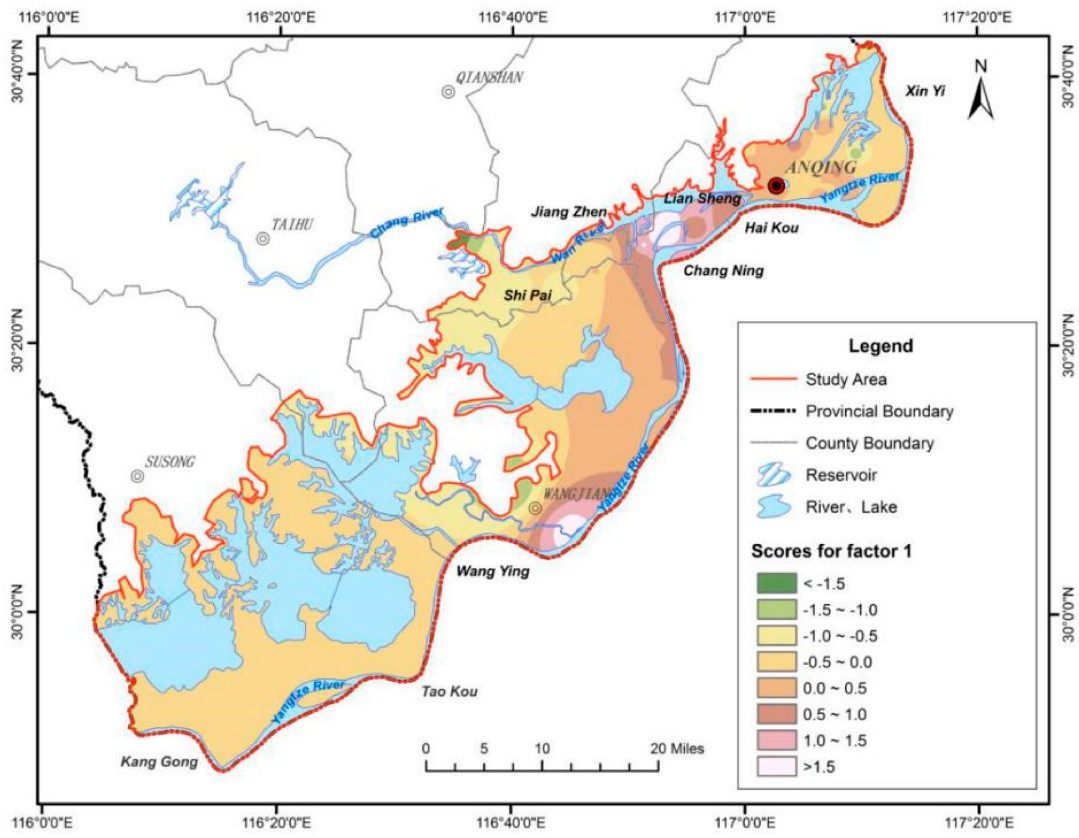

**Figure 10.** Distribution of scores of factor 1 for confined water.

In $F_2$, the factor loads of Fe, Mn, As, and $NO_3^-$ were large, and the contribution rate of $F_2$ was 22.683%. The contents of Fe, Mn, and As in confined water generally exceed the standard because of the reduction and dissolution of original iron and manganese minerals

in the aquifer and the release of arsenic in the lattice. Figure 11 shows the interpolation of $F_2$ scores at each sampling point of confined water. Sampling points with high scores are mainly distributed in the Wan River Valley plain area, where the aquifer lies at great depth and the groundwater is in a reducing environment. The dissolution of iron and manganese minerals results in the release of arsenic in the lattice. Thus, groundwater quality is controlled by the high correlation between iron, manganese, and arsenic. Moreover, this area is a crop planting area; subject to the application of agricultural nitrogen fertilizers, the content of $NO_3^-$ is high.

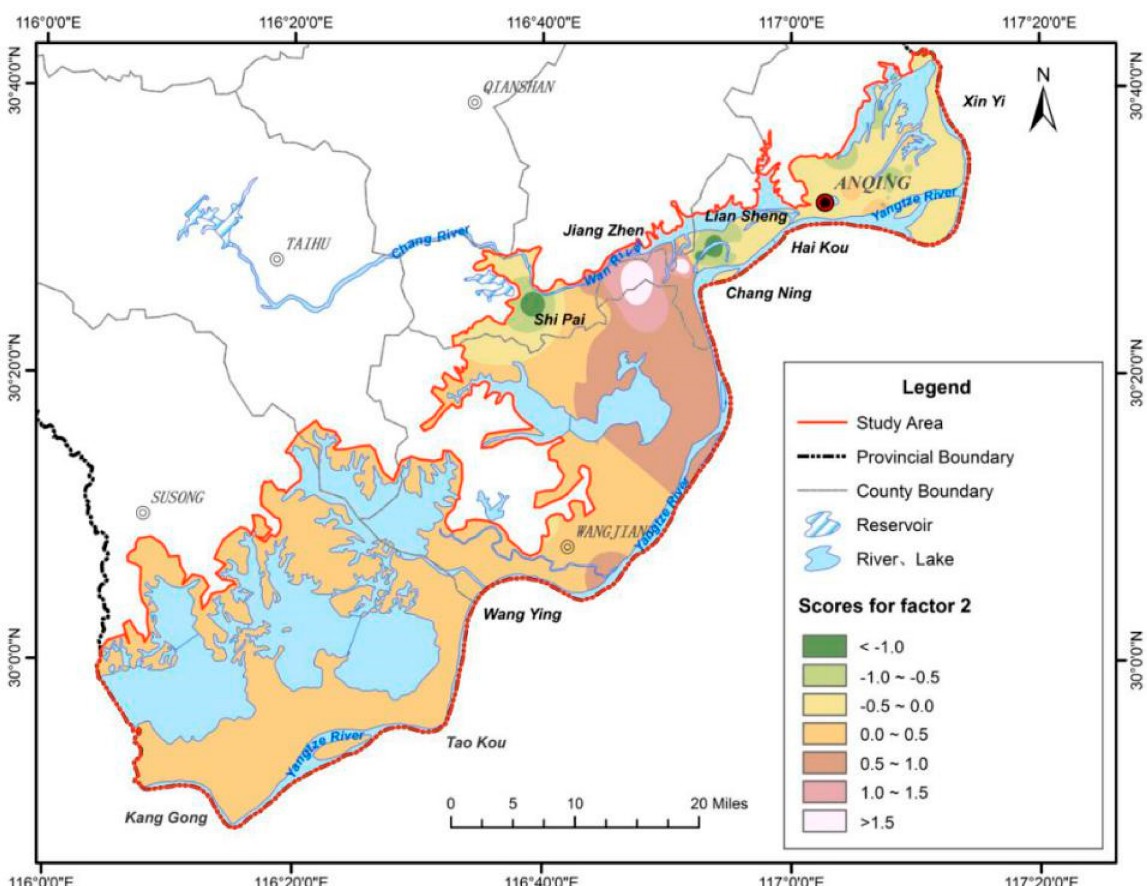

**Figure 11.** Distribution of scores of factor 2 for confined water.

$F_3$ reflects the effect of leaching of confined water. The loads of $Cl^-$ and $SO_4^{2-}$ in $F_3$ were large, and the contribution rate of $F_3$ was 17.371%. Figure 12 shows the interpolation of $F_3$ scores at each sampling point of confined water. Sampling points with high scores were mainly distributed in some areas of the riverside plain along the Anqing urban area and the Wan River Valley plain, indicating that the groundwater chemistry in this area is significantly affected by the dissolution of halite and sulfate.

$F_4$ reflects the effect of cation exchange on confined water. In $F_4$, the factor loads of $Na^+$ and $K^+$ were large, and the contribution rate of $F_4$ was 15.209%. Figure 13 shows the interpolation of $F_4$ scores at each sampling point of confined water. Sampling points with high scores were mainly distributed in the riverside plain of the Wangjiang section and the local areas of Wan River Valley plain, which indicates that halite dissolution and cation exchange are the main controlling factors, and the aquifers have a small grain size. Therefore, cation exchange is more likely to occur under such geological conditions.

Factor analysis of confined water showed that the chemistry of confined water in the study area is mainly affected by carbonate dissolution, primary pollution of iron and manganese, halite dissolution, sulfate dissolution, and cation exchange.

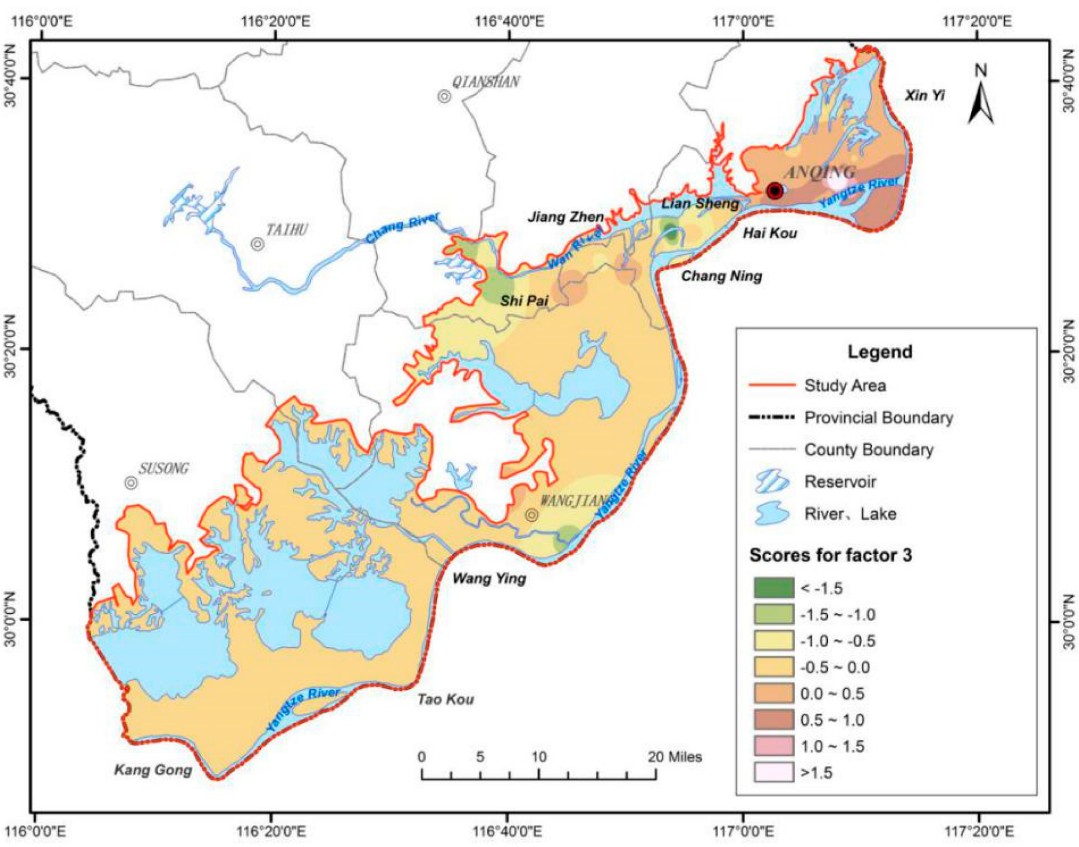

**Figure 12.** Distribution of scores of factor 3 for confined water.

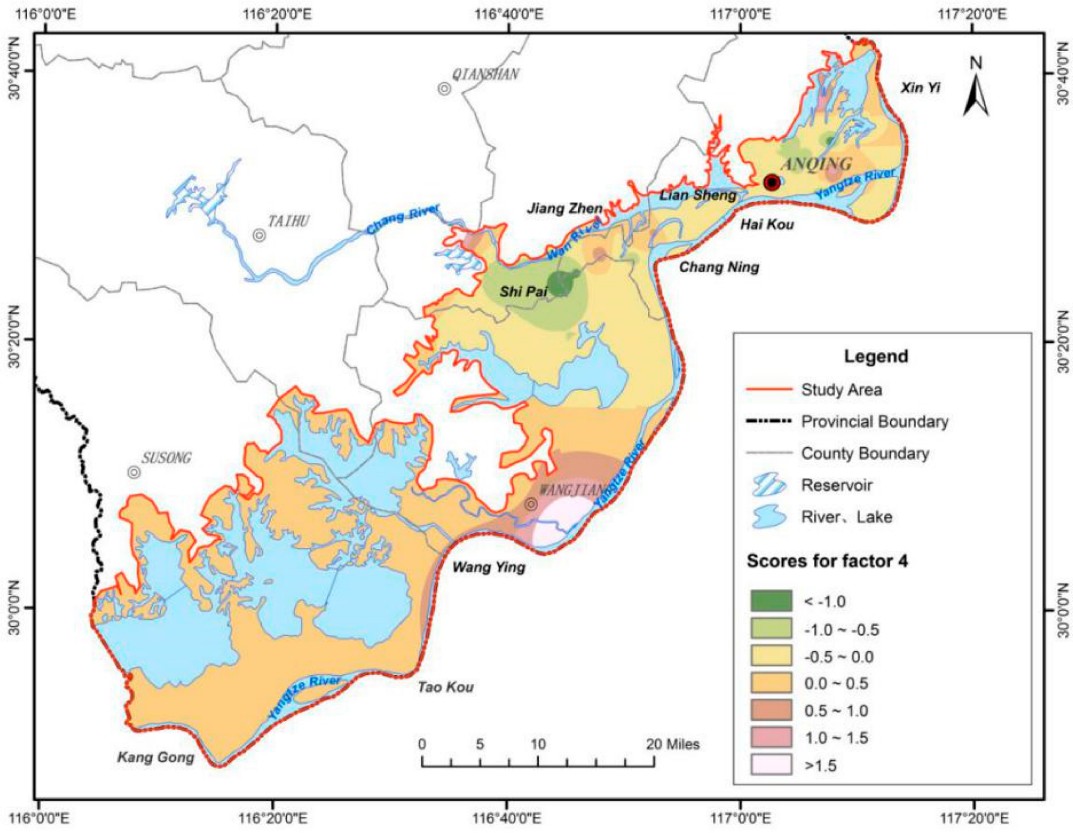

**Figure 13.** Distribution of scores of factor 4 for confined water.

### 4.3. Quantitative Analysis of Hydrochemical Evolution Mechanism: Inverse Modeling

4.3.1. Path of Simulation

The flow path of the Wan River Valley plain section (B–B′) was selected for simulation, as shown in Figure 3. According to the groundwater flow field and hydrogeological conditions in the Wan River Valley plain, groundwater flows from Wan River to the confluence of Wan River and Yangtze River. Reverse hydrogeochemical simulations were performed for the flow paths of phreatic water and confined water. The reverse hydrogeochemical simulation path of phreatic water was 45→44→40, and that of confined water was 46→42→41 (Figure 3).

4.3.2. Possible Mineral Phases

Excluding the influence of mixing, the concentration of $HCO_3^-$, $Na^+$, $Ca^{2+}$, $Mg^{2+}$, $Cl^-$ and $NO_3^-$ in groundwater increases due to the influence of water-rock interaction. Further analysis using the ion ratio method showed that water–rock interactions driving this phenomenon mainly occur as the dissolution of halites, sulfates and carbonates, and cation exchange. According to the scanning electron microscope results of the aqueous medium (Figure S1), typical iron-bearing minerals in the study area include hematite, siderite, and pyrite. Based on the above results, the main possible mineral phases can be determined as follows: calcite ($CaCO_3$), dolomite ($CaMg(CO_3)_2$), albite ($Na_2O \cdot Al_2O_3 \cdot 6SiO_2$), anorthite ($CaO \cdot Al_2O_3 \cdot 2SiO_2$), siderite ($FeCO_3$), fluorite ($CaF_2$), gypsum ($CaSO_4 \cdot H_2O$), halite ($NaCl$), hematite ($Fe_2O_3$), pyrite ($FeS_2$), and claudetite ($As_2O_3$). The wateq4f database was used for this simulation. Table 5 shows the variation of groundwater chemical components along the two flow paths.

**Table 5.** Test results of major hydrochemical components in simulated paths of phreatic water and confined water.

| Sample ID | $Na^+$ | $Ca^{2+}$ | $Mg^{2+}$ | $Cl^-$ | $SO_4^{2-}$ | $HCO_3^-$ | $F^-$ | Fe | As |
|---|---|---|---|---|---|---|---|---|---|
| | | | Phreatic water simulation path | | | | | | |
| 45 | 66.35 | 19.57 | 12.52 | 15.43 | 8.28 | 215.84 | 0.61 | 0.10 | 0.003 |
| 44 | 54.00 | 34.59 | 27.21 | 7.54 | 7.14 | 350.01 | 0.84 | 20.40 | 74.88 |
| 40 | 66.45 | 70.36 | 36.68 | 8.34 | 7.25 | 601.44 | 0.72 | 10.15 | 94.07 |
| | | | Confined water simulation path | | | | | | |
| 46 | 62.51 | 2.73 | 23.90 | 8.11 | 7.23 | 300.43 | 0.93 | 7.50 | 21.02 |
| 42 | 65.17 | 59.98 | 34.41 | 8.39 | 7.40 | 455.01 | 0.82 | 4.95 | 0.97 |
| 41 | 71.63 | 61.42 | 32.51 | 12.46 | 7.17 | 473.68 | 0.55 | 8.65 | 19.82 |

Except for As, which is in $\mu g \cdot L^{-1}$, the mass concentrations of other ions and indicators are in $mg \cdot L^{-1}$.

4.3.3. Inverse Modeling Results

Through the ion component balance calculation of groundwater samples, the saturation index (*SI*) of each mineral can be obtained to further determine the occurrence of groundwater leaching. *SI* can be expressed as follows:

$$SI = \log \frac{IAP}{K}$$

In the formula, *IAP* represents the ion activity product of the mineral components of water (dimensionless); *K* is the equilibrium constant reflected by the dissolution of minerals at a certain temperature (dimensionless).

When *SI* > 0, the mineral is supersaturated relative to the aqueous solution; when *SI* = 0, the mineral is in equilibrium with the aqueous solution; when *SI* < 0, the mineral does not reach the saturation state and will dissolve. However, the mineral saturation index remains uncertain, attributable to the errors in water quality analysis and the calculation of mineral equilibrium constant and ionic activity. Therefore, in practice, the mineral is generally considered to be in equilibrium with the aqueous solution when SI = −0.5–0.5.

According to the calculation results (Table 6), water–rock interaction occurs in the study area. In the phreatic water flow path, dolomite and hematite are in the supersaturated state and may precipitate; fluorite, gypsum, rock salt, white arsenite, and $CO_2$ (g) are unsaturated and continue to dissolve. Siderite is close to equilibrium. On the flow path of confined water, the saturation state of each mineral is consistent with that of phreatic water as a whole, while siderite is dissolved in a more reducing environment.

**Table 6.** Major mineral saturation indices along the simulated path.

| Sample ID | Dolomite | Siderite | Fluorite | Gypsum | Halite | Hematite | Claudetite | $CO_2$(g) |
|---|---|---|---|---|---|---|---|---|
| | | | Phreatic water simulation path | | | | | |
| 45 | 0.55 | −2.04 | −1.87 | −3.13 | −7.54 | 17.76 | −36.33 | −2.86 |
| 44 | 2.06 | −0.29 | −1.44 | −3.07 | −7.96 | 22.32 | −30.37 | −3.04 |
| 40 | 3.16 | −0.76 | −1.34 | −2.84 | −7.83 | 21.65 | −37.55 | −3.00 |
| | | | Confined water simulation path | | | | | |
| 46 | 0.59 | −0.45 | −2.40 | −4.07 | −7.85 | 21.49 | −36.22 | −2.93 |
| 42 | 2.82 | −1.09 | −1.24 | −2.86 | −7.83 | 21.04 | −41.19 | −3.07 |
| 41 | 3.03 | −1.12 | −1.61 | −2.88 | −7.62 | 21.47 | −39.52 | −3.19 |

In the simulation path of phreatic water, the increase in $Ca^{2+}$, $Mg^{2+}$, and $HCO_3^-$ concentrations are mainly attributable to the dissolution of calcite and dolomite, and their total dissolved amounts were 0.7582 mmol·$L^{-1}$ and 1.1755 mmol·$L^{-1}$, respectively. Fluorite was dissolved first and then precipitated and its total dissolved amount was $2.905 \times 10^{-3}$ mmol·$L^{-1}$, $Ca^{2+}$ concentration was increased and $F^-$ was released at the same time. The change of $Na^+$ concentration was mainly controlled by cation exchange. The amount of dissolved NaX was 0.8615 mmol·$L^{-1}$, and the amount of precipitated CaX was 0.4308 mmol·$L^{-1}$. The concentrations of $Na^+$ and $Cl^-$ were reduced by the precipitation of halite (0.2000 mmol·$L^{-1}$). The variation of Fe content was mainly controlled by the dissolution of hematite (1.3346 mmol·$L^{-1}$) and pyrite ($4.721 \times 10^{-2}$ mmol·$L^{-1}$), and the precipitation of siderite (2.5368 mmol·$L^{-1}$). The content of As is mainly attributable to the release of As in the crystal lattice by the reduction and dissolution of hematite and pyrite, and the dissolution of claudetite (0.4571 mmol·$L^{-1}$).

In the simulation path of confined water, the increase in $Ca^{2+}$ and $Mg^{2+}$ concentration could be mainly attributed to the dissolution of calcite (0.2564 mmol·$L^{-1}$) and dolomite (0.3962 mmol·$L^{-1}$). The precipitation of fluorite ($1.000 \times 10^{-2}$ mmol·$L^{-1}$) reduced the concentrations of $Ca^{2+}$ and $F^-$. The increase in $Na^+$ concentration was mainly controlled by cation exchange. The amount of dissolved NaX was 0.8615 mmol·$L^{-1}$, and the amount of precipitated CaX was 0.4308 mmol·$L^{-1}$. The precipitation of halite (0.2000 mmol·$L^{-1}$) reduced the concentration of $Na^+$ and $Cl^-$. The variation of Fe content is mainly controlled by the dissolution of siderite with a dissolution amount of 6.0961 mmol·$L^{-1}$, and the precipitation of hematite (2.8341 mmol·$L^{-1}$) and pyrite (0.4066 mmol·$L^{-1}$). The precipitation of claudetite (0.800 mmol·$L^{-1}$) resulted in the decrease in As content. Table 7 shows the mass exchange results of possible mineral phases on the simulated paths of phreatic and confined water.

**Table 7.** Mass exchange results of water samples along simulated paths (mmol·$L^{-1}$).

| Mineral Phases | Stoichiometry | Phreatic Water Simulation Path | | Confined Water Simulation Path | |
|---|---|---|---|---|---|
| | | 45→44 | 44→40 | 46→42 | 42→41 |
| Calcite | $CaCO_3$ | - | 0.7582 | - | 0.2564 |
| Dolomite | $CaMg(CO_3)_2$ | 0.7854 | 0.3901 | 0.4328 | $-3.663 \times 10^{-2}$ |
| Siderite | $FeCO_3$ | −3.016 | 0.4792 | 6.627 | −0.5309 |
| Fluorite | $CaF_2$ | $6.060 \times 10^{-3}$ | $-3.155 \times 10^{-3}$ | $-2.891 \times 10^{-3}$ | $-7.110 \times 10^{-3}$ |
| Gypsum | $CaSO_4·2H_2O$ | −0.1703 | $6.516 \times 10^{-2}$ | 0.8854 | $-7.312 \times 10^{-2}$ |

**Table 7.** *Cont.*

| Mineral Phases | Stoichiometry | Phreatic Water Simulation Path | | Confined Water Simulation Path | |
|---|---|---|---|---|---|
| | | 45→44 | 44→40 | 46→42 | 42→41 |
| Halite | NaCl | $-0.2226$ | $2.264 \times 10^{-2}$ | $7.959 \times 10^{-3}$ | $0.1149$ |
| Hematite | $Fe_2O_3$ | $1.650$ | $-0.3154$ | $-3.115$ | $0.2809$ |
| Pyrite | $FeS_2$ | $7.921 \times 10^{-2}$ | $-3.200 \times 10^{-2}$ | $-0.4420$ | $3.536 \times 10^{-2}$ |
| Claudetite | $As_2O_3$ | $0.4570$ | $1.283 \times 10^{-4}$ | $-1.339 \times 10^{-4}$ | $1.259 \times 10^{-4}$ |
| $CO_2$ | $CO_2$ | $3.337$ | $1.869$ | $-5.259$ | $0.4838$ |
| Cation exchange | $CaX_2$ | $-0.1138$ | $-0.3170$ | $-5.783 \times 10^{-3}$ | $-9.034 \times 10^{-2}$ |
| | NaX | $0.2276$ | $0.6339$ | $1.157 \times 10^{-2}$ | $0.1807$ |

## 5. Conclusions

Hydrogeochemical processes controlling groundwater compositions in the alluvial plain (Anqing section) of the lower Yangtze River Basin were investigated by applying conventional hydrogeochemical techniques (Piper diagram and ionic ratios), statistical methods, and inverse modeling methods to hydrochemical datasets.

The abundance of dominant cations followed the order $Ca^{2+} > Na^+ > Mg^{2+} > K^+$, and that of dominant anions followed the order $HCO_3^- > SO_4^{2-} > Cl^- > NO_3^-$. In terms of hydrochemical types of groundwater, phreatic water could be mainly classified into Ca-$HCO_3$ type and Ca-Na-$HCO_3$ type, and confined water into Ca-Na-$HCO_3$ type, Ca-$HCO_3$ type, and Ca-Na-$HCO_3$-Cl type.

The source of solutes was studied by determining relationships between ion ratios, and the main hydrogeochemical processes of various ions in groundwater were determined. The results show that $Na^+$ and $K^+$ in groundwater are mainly attributable to halite dissolution and cation alternating adsorption. $Ca^{2+}$ and $Mg^{2+}$ are mainly attributable to carbonate dissolution. Moreover, sulfate dissolution occurs during runoff, in which carbonate dissolution plays a dominant role. Cation adsorption is significant in the process of groundwater runoff, mainly manifested in the adsorption of $Ca^{2+}$ and the release of $Na^+$.

Four common factors affecting the chemical composition of phreatic water in the study area were extracted through factor analysis: carbonate dissolution ($F_1$), primary contamination of aquifer media ($F_2$), halite dissolution, and evaporation-concentration ($F_3$), and human activities ($F_4$). Similarly, four common factors affecting the chemical composition of confined water were also extracted: carbonate dissolution ($F_1$), primary contamination of aquifer media ($F_2$), dissolution of halite and sulfate ($F_3$), and cation exchange ($F_4$). According to the factor scores of each sampling point, the main control range of each factor was determined.

The results of reverse hydrogeochemical simulation showed that along the flow path in a typical profile, the hydrochemical evolution of phreatic water is mainly controlled by the dissolution of calcite, dolomite, fluorite, hematite, pyrite, and claudetite, the precipitation of halite and siderite, and cation exchange (dissolution of NaX and precipitation of CaX). The hydrochemical evolution of confined water is mainly controlled by the dissolution of calcite, dolomite, gypsum, and siderite, the precipitation of fluorite, halite, hematite, pyrite, and claudetite, and cation exchange.

The regional hydrochemical evolution law of groundwater could be analyzed in a simple manner using each method, but the comprehensive application of the ion ratio, factor analysis and reverse hydrogeochemical simulation facilitated the comprehensive investigation of the regional groundwater hydrochemical characteristics and evolution law from the macro to micro scale and from the qualitative to quantitative perspectives.

These findings provide valuable information on hydrological and hydrochemical evolution processes within aquifers of the alluvial plain (Anqing section) of the lower Yangtze River Basin. This integrated approach provides deeper insight into hydrochemical and hydrological evolution processes and a reference for ground-water management where more targeted groundwater monitoring programs would be required in the future.

Furthermore, this study provides technical support for the ecological restoration of the Yangtze River Basin.

**Supplementary Materials:** The following are available online at https://www.mdpi.com/article/10.3390/w13172403/s1, Figure S1. SEM photos of aqueous media, Table S1. Chemical component concentrations of lake water and precipitation, Table S2. According to the $^{18}$O values of lake water, precipitation and samples, the mixing concentration of each ion component is calculated. It is assumed that the ion concentration is only affected by mixing.

**Author Contributions:** Conceptualization, X.S. and Q.C.; methodology, Q.C.; software, Q.C.; validation, X.S., Q.C. and S.W.; formal analysis, X.S.; investigation, Q.C., S.Z., and Y.L.; resources, X.S.; data curation, Q.C., S.Z.; writing—original draft preparation, Q.C., Q.C., S.W., S.Z. and Y.L.; writing—review and editing, X.S., Q.C. and S.W.; visualization, Q.C.; supervision, X.S.; project administration, X.S.; funding acquisition, X.S. and Y.L. All authors have read and agreed to the published version of the manuscript.

**Funding:** This research was funded by Geological Survey project of China Geological Survey, grant number DD20189250.

**Institutional Review Board Statement:** Not applicable.

**Informed Consent Statement:** Not applicable.

**Data Availability Statement:** The study did not provide any data.

**Acknowledgments:** Our deepest gratitude goes to the anonymous reviewers for their careful work and thoughtful suggestions that have helped improve this paper substantially. This work is supported by Geological Survey project of China Geological Survey with Grant DD20189250.

**Conflicts of Interest:** The authors declare no conflict of interest.

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
