# Peer review of "Hydrochemical Characteristics and Evolution of Groundwater in the Alluvial Plain (Anqing Section) of the Lower Yangtze River Basin: Multivariate Statistical and Inversion Model Analyses"

_water, doi:10.3390/w13172403_

Round 1

Reviewer 1 Report

The authors first describe the geology and hydrology of the studied area. They then use a variety of empirical techniques (Gibbs diagrams, factor analysis) to provide insight into the water compositions. They use PHREEQC modeling to understand the change in water composition from one sampling site to another.

The manuscript is quite well prepared and reads well. The paper suffers from some approximate wording (noted above) that prevents the reader to know exactly what is going on. Although the approach chosen by the authors based on empirical parameters may provide information on water rock interactions (mineral control on water composition), more straightforward and more powerful methods exist (see below).   

First of all, chemistry is in moles and not in mass. As such, as valuable they may be, Gibbs diagrams should be replaced by equivalent diagrams using moles. It is quite easier to see the correlation between Ca and alkalinity for example, due to the often invoked carbonate dissolution.

The composition data of all the samples must be available to readers (and not only averages reported in table 1). This is the basic data from which one could generate an analysis different from that of the authors. This point is compulsory.

The authors attribute to water rock interactions and concentration by evaporation the changes in water composition. They omit referring to water mixing which certainly plays a key role in the studied area. For example it is quite likely that the Na and Cl variations aree due to pixing of waters of different salinities, instead of halite dissolution, which is quite unlikely in such a geological context (alluvial plain).

The PHREEQC modeling is difficult to understand (in fact it is not clearly explained). One would expect the authors to calculate the saturation states of all the minerals from the compositions of all the water samples. From there it is easy to see what minerals can dissolve or precipitate. The authors must also indicate what data base they used.

I cannot recommend publication at this stage. There is indeed plenty of good information in this paper that deserves publication. This is why I urge the authors to present a new manuscript along the following lines:

  • Report element concentrations versus that of an element that is likely conservative upon mixing (for example Cl). See what elements depart from a simple dilution line. Try to define endmembers.
  • Water-rock interactions: departure from mixing can be due to dissolution/precipitation. This can be discussed from the calculation of mineral saturation states. Then the authors may move on to mass transfer calculation between two sampling points (which I assume they did).
  • Ion exchange is trickier. The authors may keep all their empirical parameters to assess the extend of it.

Other remarks :

  • 2.2: chemistry is in moles not in mg. l. 189 what is the "milligram equivalent ratio".
  • 191: replace silicalite by silica or even better silicates (weathering), as done in line 192.
  • 2.3: Title: replace factor analytical by factor analysis. Remove all the sentences about medicine, psychology, etc. It is too general and useless.
  • 191-192: What else could it be other that carbonates, silicates and evaporites?
  • 3.1. Generalities about PHREEQC are not necessary either. Too general. Remove this section.
  • 1 l. 243-251. Reading would be easier if these numbers were presented in a table.
  • Table 1 is useless.
  • 366. What do you mean by "excessive"? With respect to what?
  • 3.3 l. 473. What do you mean by "possible mineral phases"? Minerals likely at equilibrium with the waters, controlling the water composition?

Author Response

Dear professor,

Thank you very much for your valuable comments and great decision. After carefully reading the comments, we revised the manuscript carefully and made detailed written explanations of the revision.

Reviewer 2 Report

The manuscript “water-1321691” aims to evaluate potential hydrogeochemical processes affecting the groundwater quality in the Alluvial Plain (Anqing section) of the Lower Yangtze  River Basin by analyzing major ions of some samples. The evolution of the water bodies were studied by means the Piper and Gibbs graphs, ion ratio method, statistical analysis and the inverse geochemical modeling.

The manuscript describes proven methods to understand the hydrogeochemical processes occurring in the studied area. However, many aspects are explained in an unclear way and require further investigation.

I believe the manuscript should be published only after major revision.

Comments (P = page#/R = row#):

P3/R97

Delete “in 2019” from caption. It is already mentioned in the text.

In Figure 1 use another color for the Section Line AA’ BB’ to make the figure clearer.

P4/

The Figure2 is not clear.  Redraw it using different colors for each lithology and increasing the size of the legend since now is not readable.

P4/

The Figure3 is not clear.  Redraw it using the same colors of Figure 2 (for the same lithology) and increasing the size of the legend. Also in this case it is not readable.

P5/R139

How was the data quality assessed for Fe, Mn and As? Please insert these information in the text

P5/R 183

This section is not clear.

The unit “milligram equivalent” isn’t correct. Maybe it is milliequivalents/L (meq/L). If it is correct, change  “milligram equivalent” with “milliequivalents/L,  (meq/L)” throughout the text.

Why the ratio (Na++K+)/Cl- greater than 1 indicates the dissolution of halite? Halite have a Na/Cl ratio equal to 1.

If K-salts are dissolved (e.g. silvite, KCl) the ratio not should change reflecting the dissolution of evaporitic rocks.

It should be noted that also atmospheric input could cause an Na/Cl ratio equal to 1. In this case a salinity plot can be used to distinguish the different contributions.

 Why the ratio (Na++K+)/Cl- less than 1 indicates the dissolution of silicate minerals? Maybe the ratio should be greater than 1.

Why the ratio (Ca2++Mg2+)/(HCO3-+SO42-) ratio less than 1 indicates silicate and evaporitic rocks dissolution?

Please explain better these concepts adding references to support these affirmations.

The authors can take inspiration from the following works for ions ratios, salinity plot and interpretation of water-rock interaction processes:

Zhang, B., Zhao, D., Zhou, P., Qu, S., Liao, F., & Wang, G. (2020). Hydrochemical characteristics of groundwater and dominant water–rock interactions in the Delingha Area, Qaidam Basin, Northwest China. Water12(3), 836.

Fuoco, I., Apollaro, C., Criscuoli, A., De Rosa, R., Velizarov, S., & Figoli, A. (2021). Fluoride Polluted Groundwaters in Calabria Region (Southern Italy): Natural Source and Remediation. Water13(12), 1626.

Apollaro, C., Fuoco, I., Bloise, L., Calabrese, E., Marini, L., Vespasiano, G., & Muto, F. (2021). Geochemical Modeling of Water-Rock Interaction Processes in the Pollino National Park. Geofluids2021.

P6/227

Insert the version of the used software.

P9/33

The results obtained using each method are discussed in separate sections.  I suggest to add a brief discussion summarizing the results validated through the combination of diverse methods applied. Please rewrite the discussion taking into account previous comments.

P12/R364

“F2 reflects the primary pollution of groundwater affected by aquifer lithology.”

This sentence in not clear. Please rewritten it.

P12/R 366-368

“Groundwater in the study area frequently exhibit excessive Fe and Mn contents, which can be mainly attributed to the reduction and dissolution of original iron and manganese minerals in the aquifer

What are the original iron and manganese minerals? “Iron and manganese minerals” it’s to generic and not appropriate term  if the authors intend to explain the dissolution mechanisms and the consequent release of As. They should list them and add a reference.

P127/R 472

The choice of mineral phases used for the elaboration of the inversion model is not clear and more details are needed.  Please describe if mineralogical analyses were conducted or add a reference concerning the main mineral phases which characterize the studied area.

P18/R508

I suggest to add a brief comment summarizing the results validated through the combination diverse methods applied. Please rewrite the conclusion taking into account previous comments

P19

Add the following works in the references list:

Zhang, B., Zhao, D., Zhou, P., Qu, S., Liao, F., & Wang, G. (2020). Hydrochemical characteristics of groundwater and dominant water–rock interactions in the Delingha Area, Qaidam Basin, Northwest China. Water12(3), 836.

Apollaro, C., Fuoco, I., Bloise, L., Calabrese, E., Marini, L., Vespasiano, G., & Muto, F. (2021). Geochemical Modeling of Water-Rock Interaction Processes in the Pollino National Park. Geofluids2021.

Fuoco, I., Apollaro, C., Criscuoli, A., De Rosa, R., Velizarov, S., & Figoli, A. (2021). Fluoride Polluted Groundwaters in Calabria Region (Southern Italy): Natural Source and Remediation. Water13(12), 1626

NOTE:

  • A spell check in all text is required
  • English should be reviewed by a native speaker

Author Response

(The authors gave the same response as above.)

Reviewer 3 Report

In reviewed article the Authors concentrate mainly on chemical composition of groundwater occurring in two Quaternary aquifers located in the Yangtze River basin. The subjects of research are shallow phreatic and deeper – confined waters of alluvial plain close to Anqing City. The paper has both some good and weak points. I would include a good selection of research methods and the right arrangement of content among the first, but in my opinion the Authors paid too much attention to factor analysis (7 pages of text!).

A considerable disadvantage of the article is the analysis of the chemical composition of groundwater, based actually only on the basic major constituents plus 3 others: Fe, Mn (which are typical in Quaternary waters) and As. The analysis of arsenic concentrations is justified due to its toxic properties, but looking at the agricultural use of the research area it is a pity that the Authors did not analyze other minor or trace constituents as, e.g. NO2-, NH4+, Al, F, P. Especially that the aim of the work was also to assess the quality of water. If the aim of the study was to assess the quality of groundwater, it was not achieved. By the way, the aim of the work should be clearly emphasized in the introduction.

Some other remarks to consider:

It would be good to correlate the results of hydrogeochemical modeling using the PHREEQC program with the mineral composition of the sediments of Quternary aquifers and aquitards.

The discussion of the results is too general and should be extended.

Remarks to the text:

  • line 48: use Piper diagram (like in other parts of the text) instead of trigram;
  • lines 50-51: Authors should avoid repeating the same words in one sentence, e.g. Statistical analysis methods based on the multivariate statistical theory can be used to determine the relationship and influence among multivariables”…Change to… Multivariate statistical methods were used to….;
  • line 53: what the Authors mean when writing „…water quality factors can be described macroscopically….”? general description or others?
  • line 71: Anqing or Anging City? In different parts of the text Authors use both form;
  • line 93: lack of dot after letter l in m a.s.l ;
  • lines 132-133: can we use aquifuge in such a Quaternary formation? Maybe better to use aquitard or aquiclude?
  • line 141: put comma after word Finally,…
  • line 153: use demineralized water instead of ultrapure water;
  • line 158: should be electrolytical conductivity and oxidation-reduction potential (Eh);
  • the chapter 4.2.2. should be rewritten. Repeating the same sentence form (in terms of the…) should be avoided;
  • avoid starting the chapter by pointing to the results from tables (chapters 4.3.2 and 4.3.3). It is better to signal the problem and then to discuss the results with reference to the tables;
  • the use of English is quite good, nevertheless I would suggest to check the text by native speaker;

Notes to figures

  • Improve the quality and readability of all figures
  • Fig.1: scale should be in kilometers (km), not miles;
  • Fig.1: to small numbers of contour of the groundwater level;
  • Fig. 2, 3: descriptions are not legible, font is too small; the same in Fig. 6;

I have no remarks to used references.

Author Response

(The authors gave the same response as above.)

Round 2

Reviewer 2 Report

In the section 3.2.1. (lines 172-196) please change the units. In this discussion the molar ratio in more appropriate.

Some mistakes  are present in the text. A text control and editing is needed before publication.

Author Response

Response to Reviewer 2

Title:Hydrochemical Characteristics and Evolution of Groundwater in the Alluvial Plain (Anqing section) of the Lower Yangtze River Basin: Multivariate Statistical and Inversion Model Analyses

Manuscript ID: water-1321691

Authors:Qiaohui Che, Xiaosi Su*, Shixiong Wang, Shida Zheng, Yunfeng Li

Corresponding author: Xiaosi Su (Email: suxiaosi@163.com)

27-August-2021

Dear professor,

Thank you very much for your valuable comments and great decision. After carefully reading the comments, we revised the manuscript carefully and made detailed written explanations of the revision.

Comments and Suggestions for Authors

In the section 3.2.1. (lines 172-196) please change the units. In this discussion the molar ratio in more appropriate.

Response: Thank you very much for your rigorous comments. We also think that mol/L is more appropriate for the analysis of the ratio relationship between Na+ and Cl-, C2+ and SO42-. We have modified the unit in section 3.2.1, replacing meq/L with mol/L, and modified the corresponding discussion part and pictures.

Some mistakes are present in the text. A text control and editing is needed before publication.

Response: Thank you very much for your rigorous comments. We rechecked the manuscript again and found that there were spelling errors. For example, "e" was missing in "are the main recharge sources" in 4.2.1 (line 295), and so on. We have modified them.

Sincerely,

Xiaosi Su (on behalf of all authors)

Institute of Water Resources and Environment, Jilin University, Changchun, 130012, China; College of Construction Engineering, Jilin University, Changchun 130021, China. E-mail: suxiaosi@163.com.